# Improving Graph Generation with Flow Matching and Optimal Transport

## Abstract

Generating graph-structured data is crucial in various domains but remains challenging due to the complex interdependencies between nodes and edges. While diffusion models have demonstrated their superior generative capabilities, they often suffer from unstable training and inefficient sampling. To enhance generation performance and training stability, we propose GGFlow, a discrete flow matching generative model incorporating optimal transport for graph structures and it incorporates an edge-augmented graph transformer to enable the direct communications among edges. Additionally, GGFlow introduces a novel goal-guided generation framework to control the generative trajectory of our model towards desired properties. GGFlow demonstrates superior performance on both unconditional and conditional generation tasks, outperforming existing baselines and underscoring its effectiveness and potential for wider application.

## 1 Introduction

Graph structural data generation has become critically important across various domains, including social networks (Grover et al., 2019), drug design (Bilodeau et al., 2022), and neural architecture search (NAS) (Lee et al., 2020). Effective modeling of the intrinsic joint distribution and accurate description of topological structures of graphs are essential for these applications. Deep generative models have increasingly demonstrated success in graph generation by effectively modeling the complex structural properties of graphs. These models are typically categorized into autoregressive and one-shot types. Autoregressive models, such as GraphRNN (You et al., 2018), generate graphs sequentially, often overlooking the interdependencies among all graph components. In contrast, one-shot methods generate entire graphs in a single step, more effectively capturing the joint distribution (Ma et al., 2018; Luo et al., 2023; Niu et al., 2020).

Diffusion models have shown great promise and achieved significant performance in various domains (Ho et al., 2020; Song et al., 2020; Ho et al., 2022). In the context of graph generation, diffusion models have been adopted to enhance generative capacity. EDP-GNN and GDSS are among the first to utilize diffusion models for graph generation, adding continuous Gaussian noise to adjacency matrices and node types, which may lead to invalid graph structures (Niu et al., 2020; Jo et al., 2022b). Due to the inherent sparsity and discreteness of graph structures, GSDM enhances model fidelity by introducing Gaussian noise within a continuous spectrum space of the graph, and DiGress and PPGN apply discrete diffusion models for graphs (Luo et al., 2023; Vignac et al., 2022; Austin et al., 2021; Haefeli et al., 2022; Huang et al., 2023).

Despite their potential, diffusion models often face challenges with unstable training and inefficient sampling. Flow matching generative models offer a more stable and efficient alternative by transforming the generative process from stochastic differential equations (SDEs) to ordinary differential equations (ODEs), enhancing generative efficiency (Lipman et al., 2022; Song et al., 2024; Yim et al., 2023). Additionally, the use of optimal transport (OT) straightens the marginal probability path, reducing training variance and speeding up sampling (Bose et al., 2023; Tong et al., 2023; Klein et al., 2024; Pooladian et al., 2023). While the application of OT in graph-based systems is often hampered by significant computational demands, primarily due to the complexity of the OT metric (Chen et al., 2020b; Petric Maretic et al., 2019).

In this paper, we introduce GGFlow, a novel generative model that leverages discrete flow matching techniques with optimal transport to improve sampling efficiency and training stability in graph

generation. The model preserves graph sparsity and permutation invariance, which is essential for realistic graph generation. Additionally, GGFlow employs a goal-guided framework using reinforcement learning for conditional generation. GGFlow achieves state-of-the-art results in both unconditional and conditional graph and molecule generation tasks and surpasses existing methods with fewer inference steps. Its effectiveness in conditional generation tasks underscores the practical impact of our approach.

Our contribution can be summarized as:

- GGFlow introduces the first discrete flow matching generative model with optimal transport for graph data, improving sampling efficiency and training stability. It also incorporates an edge-augmented graph transformer to enhance generation tasks further.

- GGFlow proposes a novel guidance framework using reinforcement learning to control probability flow during graph generation, targeting specific properties.

- GGFlow demonstrates state-of-the-art performance in various unconditional and conditional graph generation tasks, consistently outperforming existing methods across diverse graph types and complexities.

## 2 RELATED WORK

### 2.1 FLOW MATCHING AND DIFFUSION MODELS

Diffusion models have gained widespread popularity in various fields, including computer vision, natural language processing, and biological sciences, demonstrating notable success in generative tasks (Ho et al., 2020; Song et al., 2020; Watson et al., 2023; Ingraham et al., 2023; Liu et al., 2024a; Ren et al., 2024; Zhu et al., 2024). However, these models often suffer from inefficiencies in sampling due to the complexity of their underlying diffusion processes and the convergence properties of the generative process.

Flow matching generative models have emerged as a more efficient and stable alternative (details in Appendix A.1), improving sampling by straightening the generative probability path (Lipman et al., 2022; Song et al., 2024; Campbell et al., 2024). Some approaches further enhance performance by incorporating optimal transport. The generative processes of these models are summarized in Figure 1.

Previous works (Campbell et al., 2024; Gat et al., 2024) extended flow matching to discrete spaces, while Eijkelboom et al. (2024) applied variational flow matching to graphs, but without adequately addressing key graph-specific properties such as adjacency matrix sparsity. GGFlow tackles these challenges by introducing a discrete flow matching model with optimal transport tailored for graph data. Furthermore, we propose a novel framework for guiding the generative process, enhancing its practical applicability.

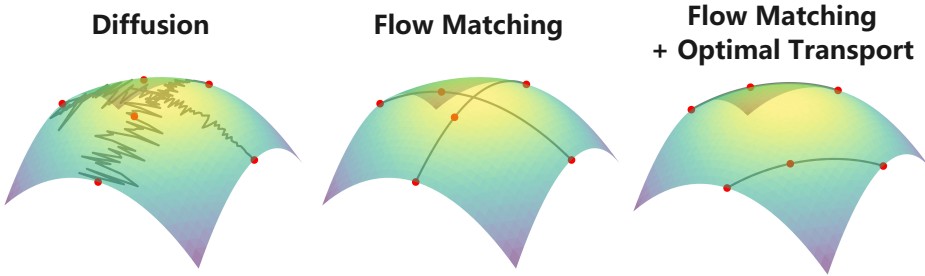

Figure 1: Illustration of generative trajectories using different methods. The generative trajectories are learned by the diffusion model (left), flow matching model (center), and flow matching model with optimal transport (right).

## 2.2 Graph Generative Models

Graph generative models are typically categorized into two main types: autoregressive and one-shot models. Autoregressive models, such as generative adversarial networks (Wang et al., 2018), recurrent neural networks (You et al., 2018), variational autoencoders (Jin et al., 2018), normalizing flows (Shi et al., 2019; Luo et al., 2021) and diffusion model (Kong et al., 2023), generate graphs sequentially. While effective, these models are often computationally expensive and fail to account for permutation invariance, a crucial property for graph data, resulting in potential inefficiencies. In contrast, one-shot models aim to capture the distribution of all graph components simultaneously (De Cao and Kipf, 2018; Ma et al., 2018; Zang and Wang, 2020), better reflecting the inherent interactions within graphs. Despite the advantages, diffusion-based one-shot models (Niu et al., 2020; Jo et al., 2022b; Vignac et al., 2022; Chen et al., 2023; Bergmeister et al., 2023; Luo et al., 2023; Haefeli et al., 2022; Yan et al., 2023; Jang et al., 2023; Madeira et al., 2024; Bergmeister et al., 2024; Chen et al., 2023; Minello et al., 2024; Zhao et al., 2024; Xu et al., 2024) show promising results in downstream tasks but remain limited by sampling efficiency. GGFlow addresses these limitations by employing a discrete flow-matching generative model, achieving superior generative performance with fewer sampling steps. More comparisons with recent works are presented in Appendix B.

## 3 Methods

In this section, we present our methodology, GGFlow. Section 3.1 outlines the discrete flow matching method for graph generation. Section 3.2 covers optimal transport for graph flow matching. Section 3.3 introduces GraphEvo, our neural network for graph generation. Section 3.4 examines the permutation properties of GGFlow, and Section 3.5 discusses goal-guided graph generation using reinforcement learning.

### 3.1 Discrete Flow Matching for Graph Generation

A graph $G = (V, E)$, where $V$ and $E$ denote the sets of nodes and edges, has a distribution denoted by $p(G) = (p^V(V), p^E(E))$. The attribute spaces for nodes and edges are $\mathcal{V}$ and $\mathcal{E}$, with cardinalities $n$ and $m$, respectively. The attributes of node $i$ and edge $ij$ are denoted by $v_i \in \mathcal{V}$ and $e_{ij} \in \mathcal{E}$, so the node and edge probability mass functions (PMF) are $p^V(v_i = a)$ and $p^E(e_{ij} = b)$ where $a \in \{1, \ldots, n\}$ and $b \in \{1, \ldots, m\}$. The node and edge encodings in the graph are given by matrices $\mathbf{V} \in \mathbb{R}^{a \times n}$ and $\mathbf{E} \in \mathbb{R}^{a \times a \times m}$, respectively. We denote the transpose of matrix $\mathbf{A}$ as $\mathbf{A}^*$ and $\mathbf{A}^t$ represents the state of matrix $\mathbf{A}$ at time $t$. We use discrete flow matching to model the graph generation process.

**Source and target distribution** GGFlow aims to transform prior distribution $G^0 \sim p_{\text{ref}}$ to target data distribution $G^1 \sim p_{\text{data}}$. The training data $(G^0, G^1)$ are sampled from a joint distribution $\pi(G^0, G^1)$, satisfying the marginals constraints $p_{\text{ref}} = \sum_{G^1} \pi(G^0, G^1), p_{\text{data}} = \sum_{G^0} \pi(G^0, G^1)$. In the simplest case, the joint distribution $\pi(G^0, G^1)$ is modeled as the independent coupling, i.e. $\pi(G^0, G^1) = p_{\text{ref}} \cdot p_{\text{data}}$.

To account for graph sparsity, the prior distribution $p_{\text{ref}} = (p_{\text{ref}}^V, p_{\text{ref}}^E)$ is designed to approximate the true data distribution closely. To ensure the permutation invariance of the model, the priors are structured as products of single distributions for all nodes and edges: $\prod_i v_i \times \prod_{ij} e_{ij}$ (Vignac et al., 2022). Further details on the prior can be found in Appendix C.1.

**Probability path** We define a probability path $p_t(G^t)$ that interpolates between source distribution $p_{\text{ref}}$ and target distribution $p_{\text{data}}$ i.e. $p_0 = p_{\text{ref}}$ and $p_1 = p_{\text{data}}$. The marginal probability path is given by:

$$p_t(G^t) = \sum_{(G^0, G^1) \sim \pi} p_t(G^t | G^0, G^1) \pi(G^0, G^1), \tag{1}$$

where

$$p_t(G^t | G^0, G^1) = \text{Cat}\Big(t\delta\{G^1, G\} + (1-t)p_{\text{ref}}\Big)$$
$$= \text{Cat}\Big(t\delta\{V^1, V\} + (1-t)p_{\text{ref}}^V, t\delta\{E^1, E\} + (1-t)p_{\text{ref}}^E\Big),$$

$\delta$ is the Kronecker delta, indicating equality of the indices, and $\text{Cat}(p)$ denotes a Categorical distribution with probabilities $p$. Given the sparsity of both the prior and data distributions, we can infer that the intermediate distribution is similarly sparse, aiding model training.

We define a probability velocity field $u_t(G, G^t) = (u_t^V(V, V^t), u_t^E(E, E^t))$ for GGFlow, which generates the probability path from Equation 1. The probability velocity field $u_t(G, G^t)$ is derived from the conditional probability velocity field $u_t(G, G^t | G^0, G^1)$, and can be expressed as:

$$u_t(G, G^t) = \sum_{(G^0, G^1) \sim \pi} u_t(G, G^t | G^0, G^1) p_t(G^0, G^1 | G^t), \tag{2}$$

$$p_t(G^0, G^1 | G^t) = p_{1|t}(G^1 | G^t, G^0) \frac{p_t(G^t | G^0, G^1) \pi(G^0, G^1)}{\sum_{G^0, G^1} p_t(G^t | G^0, G^1) \pi(G^0, G^1)}. \tag{3}$$

GGFlow chooses the conditional marginal probability $u_t(G, G^t | G^0, G^1)$ as:

$$u_t(G, G^t | G^0, G^1) = \frac{1}{\mathbf{Z}_t (1-t) p_{\text{ref}}} \delta\{G, G^1\} (1 - \delta\{G^t, G^1\}), G_t \neq G, \tag{4}$$

where $\text{ReLU}(a) = \max(a, 0)$ and $\mathbf{Z}_t = |\{G^t : p_t(G^t | G^0, G^1) > 0\}|$. More details about the conditional vector field are provided in Appendix C.2.

**Training objective** Given the intractability of the posterior distribution $p_{1|t}(G^1 | G^t, G^0)$, we approximate it as $\hat{p}_{1|t}(G^1 | G^t, G^0)$ using neural network, as detailed in Section 3.3. The training objective is formulated as:

$$\mathcal{L} = \mathbb{E}_{p_{\text{data}}(G^1) \mathcal{U}(t; 0, 1) \pi(G^0, G^1) p_t(G^t | G^0, G^1)} [\log \hat{p}_{1|t}(G^1 | G^t, G^0)], \tag{5}$$

where $\mathcal{U}(t; 0, 1)$ is a uniform distribution on $[0, 1]$.

**Sampling Procedure** In the absence of the data distribution $G^1$ during sampling, we reparameterize the conditional probability $p_t(G^0, G^1 | G^t)$ as:

$$p_t(G^0, G^1 | G^t) = p_{1|t}(G^1 | G^t, G^0) \frac{p_t(G^t | G^0) p(G^0)}{\sum_{G^0} p_t(G^t | G^0) p(G^0)}.$$

$$p_t(G^t | G^0) = \text{Cat}\Big( t\delta\{V^1, V\} + (1-t) p_V^{\text{ref}}, t\delta\{E^1, E\} + (1-t) p_E^{\text{ref}} \Big)$$

And we can simplify the generative process $p_{t+\Delta t|t}(G^{t+\Delta t} | G^t, G^0)$ without the calculation of the full expectation over conditional vector field $u_t(G, G^t | G^0, G^1)$:

$$p_{t+\Delta t|t}(G^{t+\Delta t} | G^t, G^0) = \mathbb{E}_{\hat{p}_t(G^1 | G^t, G^0)} [\delta(G^t, G^{t+\Delta t}) + u_t(G^t, G^{t+\Delta t} | G^0, G^1) \Delta t]$$

$$= \sum_{G^1} p_{t+\Delta t|t}(G^{t+\Delta t} | G^1, G^t, G^0) \hat{p}_{1|t}(G^1 | G^t, G^0). \tag{6}$$

We first sample the $\hat{G}^1$ using the approximate distribution $\hat{p}_{1|t}(G^1 | G^t, G^0)$ and then sample the next state $G^{t+\Delta t}$ using sampled $\hat{G}^1$. The sampling procedure $p_{t+\Delta t|t}(G^{t+\Delta t} | G^1, G^t, G^0)$ can thus be formulated as:

$$G^{t+\Delta t} \sim \delta\{\cdot, G^t\} + u_t(\cdot, G^t | G^0, \hat{G}^1) \Delta t.$$

Further details on the sampling and training procedures are provided in Algorithms 1 and 4.

## 3.2 OPTIMAL TRANSPORT FOR GRAPH FLOW MATCHING

Optimal transport (OT) has been effectively applied to flow matching generative models in continuous variable spaces, to improve generative performance (Tong et al., 2023; Bose et al., 2023; Song et al., 2024). To generalize this for graphs, we extend the joint distribution $\pi(G^0, G^1)$ from independent coupling to the 2-Wasserstein OT map $\phi^*$, which minimizes the 2-Wasserstein distance between $p_{\text{ref}}$ and $p_{\text{data}}$. To optimize the computational efficiency of OT, we define the distance via the Hamming distance $H(G^1, G^0)$ (Bookstein et al., 2002):

$$\phi^*(p_0, p_1) = \arg\inf_{\phi \in \Phi} \int_{\mathbb{R}^d \times \mathbb{R}^d} H(G^0, G^1) \mathrm{d}\phi(G^0, G^1), \tag{7}$$

---

**Algorithm 1** Sampling Procedure of GGFlow

---

**Require:** $t = 0, G^0 \sim (p_V^{\text{ref}}, p_E^{\text{ref}}), u_t(G, G^t | G^0, G^1), N_{\text{steps}}$
1: $\Delta t = 1/N_{\text{steps}}$
2: **for** $n \in \{0, \ldots, N_{\text{steps}} - 1\}$ **do**
3: $\quad \hat{p}_{1|t}(G^1 | G^0, G^t) = \text{GraphEvo}(G^t, G^0, t)$
4: $\quad \hat{G}^1 \sim \hat{p}_{1|t}(\cdot | G^0, G^t)$
5: $\quad$ // Sampling from the conditional velocity field
6: $\quad G^{t+\Delta t} \sim \delta\{\cdot, G^t\} + u_t(\cdot, G^t | G^0, \hat{G}^1)\Delta t$
7: $\quad t = t + \Delta t$
8: **end for**
9: **return** $G^1 = (V^1, E^1)$

---

where

$$H(G^0, G^1) = \sum_i \delta(v_i^0, v_i^1) + \lambda \sum_{i,j} \delta(e_{ij}^0, e_{ij}^1). \tag{8}$$

Here $\Phi$ represents the set of all joint probability measures on $\mathbb{R}^d \times \mathbb{R}^d$ that are consistent with the marginal distributions $p_0$ and $p_1$, where $G^K = (V^K = \{v_i^K\}, E^K = \{e_{ij}^K\}_{ij}), K = 0, 1$.

The practical application of OT to large datasets is computationally intensive, often requiring cubic time complexity and quadratic memory (Tong et al., 2020; Villani, 2009). To address these challenges, we use a minibatch approximation of OT (Fatras et al., 2021). A detailed time analysis of optimal transport during the training procedure is provided in Appendix E..

### 3.3 GRAPHEVO: EDGE-AUGMENTED GRAPH TRANSFORMER

Our neural network, GraphEvo, predicts the posterior distribution $\hat{p}_{1|t}(G^1 | G^t, G^0)$ using the intermediate graph $G^t$ and initial noise graph $G^0$. In graph-structured data, edge and structural information are as critical as node attributes, and incorporating edge relations enhances link generation tasks (Hussain et al., 2024; Hou et al., 2024; Jumper et al., 2021). To capture these relations, GraphEvo extends the graph transformer by introducing a triangle attention mechanism for edge updates, along with additional graph features $y$, such as cycles and the number of connected components (Vignac et al., 2022). This enables GraphEvo to efficiently and accurately capture the joint distribution of all graph components. The key self-attention mechanisms are outlined in Algorithm 2, where node, edge, and graph features are represented as $\mathbf{X} \in \mathbb{R}^{bs \times n \times dx}$, $\mathbf{E} \in \mathbb{R}^{bs \times n \times dx}$, and $\mathbf{y} \in \mathbb{R}^{bs \times n \times dy}$, where $bs$ denotes batch size, $n$ is the number of nodes, and $dx$ and $dy$ are the feature dimensions for node and global features, respectively. Further details are provided in Appendix D.

---

**Algorithm 2** Self-attention Mechanism in GraphEvo

---

**Require:** $\mathbf{X} \in \mathbb{R}^{bs \times n \times dx}, \mathbf{E} \in \mathbb{R}^{bs \times n \times dx}, \mathbf{y} \in \mathbb{R}^{bs \times n \times dy}$
1: $\mathbf{Q}, \mathbf{K}, \mathbf{V} \leftarrow \text{Linear}(\mathbf{X})$
2: $\mathbf{Y} \leftarrow \frac{\mathbf{Q} \times \mathbf{K}}{\sqrt{d_{\mathbf{Y}}}}$ $\quad$ // Calculation attention score for node embedding
3: $\mathbf{Y} \leftarrow \text{FiLM}(\mathbf{Y}, \mathbf{E})$ $\quad$ // Incorporate edge features to self-attention scores
4: $\mathbf{E} \leftarrow \mathbf{Y}$
5: $\mathbf{Q_e}, \mathbf{K_e}, \mathbf{V_e}, \mathbf{b}, \mathbf{g} \leftarrow \text{Linear}(\mathbf{E})$
6: $\mathbf{Y_e} \leftarrow \frac{\mathbf{Q_e} \times \mathbf{K_e}}{\sqrt{d_{\mathbf{Y_e}}}} + \mathbf{b}$ $\quad$ // Calculation triangle attention score for edge embedding
7: $\mathbf{E} \leftarrow \mathbf{Y_e} * \mathbf{V_e} * \text{sigmoid}(\mathbf{g})$
8: $\mathbf{E} \leftarrow \text{Linear}\Big(\text{FiLM}(\mathbf{E}, \mathbf{y})\Big)$ $\quad$ // Incorporate global structural features to edge embedding
9: $\mathbf{X} \leftarrow \mathbf{Y} * \mathbf{V}$
10: $\mathbf{X} \leftarrow \text{Linear}\Big(\text{FiLM}(\mathbf{X}, \mathbf{y})\Big)$ $\quad$ // Incorporate global structural features to node embedding
11: $\mathbf{y} \leftarrow \text{Linear}\Big(\text{Linear}(\mathbf{y}) + \text{PNA}(\mathbf{X}) + \text{PNA}(\mathbf{E})\Big)$
12: **return** $\mathbf{X}, \mathbf{E}, \mathbf{y}$

---

### 3.4 PERMUTATION PROPERTY ANALYSIS

Graphs are invariant to random node permutations, and GGFlow preserves this property. To ensure permutation invariance, we analyze the permutation properties of our neural network, training objectives, and conditional probabilities path. First, we analyze the permutation invariance of the training objectives (Vignac et al., 2022). Since the source and target distributions are permutation invariant, the independent coupling also exhibits this invariance. Our optimal transport map, derived from Equation 7, similarly demonstrates invariance to identical permutations. Further clarifications regarding optimal transport can be found in Appendix C.4.

**Theorem 1.** *If the distributions $p(G^0)$ and $p(G^1)$ are permutation invariant, and the cost function maintains invariance under identical permutations, i.e., $H(G^0, G^1) = H(\pi G^0, \pi G^1)$ for any permutation $\pi$, then the optimal transport map $\phi$ also exhibits invariance under identical permutations, such that $\phi(G^0, G^1) = \phi(\pi G^0, \pi G^1)$.*

Proof of this theorem can be found in Appendix C.4. To ensure that the generated graph retains its identity under random permutations, the generated distribution must remain exchangeable, and GraphEvo must be permutation equivariant.

**Proposition 1.** *The distribution generated by the conditional flow is exchangeable with respect to nodes and graphs, i.e. $p(\mathbf{V}, \mathbf{E}) = \mathbf{p}(\pi^* \mathbf{V}, \pi^* \mathbf{E}\pi)$, where $\pi$ is a permutation operator.*

**Proposition 2.** *GraphEvo is permutation equivariant.*

The proofs of Proposition 1 and 2 are provided in Appendix C.3 and Appendix D.1, respectively.

### 3.5 GOAL-GUIDED FRAMEWORK FOR CONDITIONAL GENERATION

For practical applications such as drug discovery, we propose a goal-guided framework for discrete flow matching, employing reinforcement learning (RL) to guide graph flow matching models for non-differentiable objectives. The goal of the guidance method is to map the noise distribution $p_0$ to a preference data distribution $p_1^*$ using a reward function $\mathcal{R}(G^t, t)$.

We formulate the inference process of flow matching as a Markov Decision Process (MDP), where $(G^t, t)$ and $G^{t+\Delta t}$ are the state space $\mathbf{s}_t$ and action space $\mathbf{a}_t$, $p_0$ is an initial noise distribution, $p_{t+\Delta t|t}(G^{t+\Delta t}|G^t, t)$ is the transition dynamics and policy network $\pi(\mathbf{a}_t|\mathbf{s}_t)$, $\mathcal{R}(G^t, t) = r(G^1)\mathbb{I}[t=1]$ is the reward function

To enable exploration, we introduce a temperature parameter $T$ for the policy network during sampling, allowing the model to explore a broader space at higher temperatures:

$$\pi(\mathbf{a}_t|\mathbf{s}_t) = p_{t+\Delta t|t}(G^{t+\Delta t}|G^t, t) = \mathrm{Cat}\Big((\delta\{\cdot, G^t\} + u_t(\cdot, G^t|G^0, \hat{G}^1)\Delta t)/T\Big) \quad (9)$$

The goal of RL training is to maximize the reward function. To prevent overfitting to the reward preference distribution, we add a Kullback–Leibler (KL) divergence term between the Reinforcement learning fine-tuned model $p_\theta^{RL}(\cdot)$ and pre-trained model $p_\theta(\cdot)$ (Ouyang et al., 2022).

We employ the policy gradient method to update the network, where the policy is refined to $\pi(\mathbf{a}_t|\mathbf{s}_t) = p_\theta^{(T)}(G^1|G^t)q(G^{t+\Delta t}|G^1)$ to $\pi(\mathbf{a}_t|\mathbf{s}_t) = p_\theta^{(T)}(G^1|G^t)$ (Sutton et al., 1999; Liu et al., 2024b), directly increasing the probability of generating $G^1$ with higher rewards at all timestep $t$. The training objective is:

$$\mathcal{L}_{RL} = -\mathbb{E}_{p_\theta(G^{0:t:1})}[\alpha\mathcal{R}(G^1)\sum_{t=0}^{t=1}\log p_\theta^{RL}(G^1|G^t, G^0) - \beta\sum_{t=0}^{t=1}\mathrm{KL}(p_\theta^{RL}(G^1|G^t, G^0)||p_\theta(G^1|G^t, G^0))]$$

$$(10)$$

where $p_\theta(G^{0:t:1})$ represents $p_{\mathrm{data}}(G^1)\mathcal{U}(t; 0, 1)\pi(G^0, G^1)p_t(G^t|G^0, G^1)$. Using this optimization objective, we fine-tune the pre-trained flow matching model to generate data following the preference distribution. By integrating optimal transport, we optimize the pairing of prior data and high-reward training data (Chen et al., 2020a). The pseudo-code for the guided GGFlow training is provided in Algorithm 5 and a toy example is shown in Appendix G.

## 4 EXPERIMENT

To validate the performance of our method, we compare GGFlow with state-of-the-art graph generative baselines on generic graph generation and molecule generation, over several benchmarks in Section 4.1 and Section 4.2, respectively. The ability of GGFlow to perform conditional generation is analyzed in Section 4.3. Finally, we conduct detailed ablation studies presented in Section 4.4.

### 4.1 GENERIC GRAPH GENERATION

We evaluated GGFlow on five generic graph generation benchmarks of varying sizes: Ego-small, Community-small, Grid, Planar and Enzymes. We employ the same train/test split as GraphRNN (You et al., 2018), utilizing 80% of each dataset for training and the remaining for testing. We compared GGFlow's performance against well-known autoregressive models: DeepGMG (Li et al., 2018), GraphRNN (You et al., 2018), GraphAF (Shi et al., 2019), and GraphDF (Luo et al., 2021) and one-shot models: GraphVAE (Simonovsky and Komodakis, 2018), GNF (Liu et al., 2019), EDP-GNN (Niu et al., 2020), GDSS (Jo et al., 2022a), DiGress (Vignac et al., 2022), GRASP (Minello et al., 2024), GSDM (Luo et al., 2023), GruM (Jo et al., 2024), and SwinGNN (Yan et al., 2023). Consistent with previous studies, we generated an equal number of graphs as the test set to compare distributions of graph statistics, including degree distribution (Deg.), clustering coefficient (Clus.), and the frequency of 4 node orbits (Orbit). Detailed descriptions of datasets, baselines, and metrics are provided in Appendix I.

Table 1 presents our results, showing that GGFlow achieves superior performance across most metrics. Additionally, GGFlow demonstrates comparable performance compared to state-of-the-art models in generating large graphs on the Grid dataset. These findings underscore the effectiveness of GGFlow at capturing the local characteristics and data distributions of graphs. Additional metrics and dataset experimental results are included in Appendix H, and we visualize the generated graphs in Appendix K.

Table 1: Generation results on the generic graph datasets. Results are the means of 3 different runs. The best results and the second-best results are marked **bold** and bold.

| Method | Ego-small | | | | Community-small | | | | Grid | | | | Step |
|---|---|---|---|---|---|---|---|---|---|---|---|---|---|
| | Deg. | Clus. | Orbit | Avg. | Deg. | Clus. | Orbit | Avg. | Deg. | Clus. | Orbit | Avg. | |
| Training Set | 0.014 | 0.022 | 0.004 | 0.013 | 0.003 | 0.009 | 0.001 | 0.005 | 0.000 | 0.000 | 0.000 | 0.000 | - |
| DeepGMG | 0.040 | 0.100 | 0.020 | 0.053 | 0.220 | 0.950 | 0.400 | 0.523 | - | - | - | - | - |
| GraphRNN | 0.090 | 0.220 | 0.003 | 0.104 | 0.080 | 0.120 | 0.040 | 0.080 | 0.064 | 0.043 | 0.021 | 0.043 | - |
| GraphAF | 0.031 | 0.107 | **0.001** | 0.046 | 0.178 | 0.204 | 0.022 | 0.135 | - | - | - | - | - |
| GraphDF | 0.039 | 0.128 | 0.012 | 0.046 | 0.060 | 0.116 | 0.030 | 0.069 | - | - | - | - | - |
| GNF | 0.030 | 0.100 | **0.001** | 0.044 | 0.200 | 0.200 | 0.110 | 0.170 | - | - | - | - | - |
| GraphVAE | 0.137 | 0.166 | 0.051 | 0.118 | 0.358 | 0.969 | 0.551 | 0.626 | 1.594 | **0.000** | 0.904 | 0.833 | - |
| EDP-GNN | 0.054 | 0.092 | 0.007 | 0.051 | 0.050 | 0.159 | 0.027 | 0.079 | 0.460 | 0.243 | 0.316 | 0.340 | 1000 |
| GDSS | 0.027 | **0.033** | 0.008 | 0.022 | 0.044 | 0.098 | 0.009 | 0.058 | 0.133 | 0.009 | 0.123 | 0.088 | 1000 |
| GSDM | - | - | - | - | 0.020 | 0.050 | 0.005 | 0.053 | 0.002 | **0.000** | **0.000** | 0.001 | 1000 |
| DiGress | 0.028 | 0.046 | 0.008 | 0.027 | 0.032 | 0.047 | 0.009 | 0.025 | 0.037 | 0.046 | 0.069 | 0.051 | 500 |
| SwinGNN | 0.017 | 0.060 | 0.003 | 0.027 | **0.006** | 0.125 | 0.018 | 0.050 | **0.000** | **0.000** | **0.000** | **0.000** | 500 |
| GGFlow | **0.005** | **0.033** | 0.004 | **0.014** | 0.011 | **0.030** | **0.002** | **0.014** | 0.030 | **0.000** | 0.016 | 0.015 | 500 |

### 4.2 MOLEUCLE GRAPH GENERATION

We evaluated GGFlow on two standard molecular datasets, QM9 (Ramakrishnan et al., 2014) and ZINC250k (Irwin et al., 2012), using several metrics: Validity, Validity without correction, Neighborhood Subgraph Pairwise Distance Kernel (NSPDK) Maximum Mean Discrepancy (MMD), and Frechet ChemNet Distance (FCD). To calculate these metrics, we sampled 10,000 molecules. We compared GGFlow against various molecule generation models, including GraphAF, GraphDF, MolFlow (Zang and Wang, 2020), EDP-GNN, GraphEBM (Liu et al., 2021), GDSS, PS-VAE (Kong et al., 2022), MolHF (Zhu et al., 2023), GruM, SwinGNN, DiGress, and GSDM. Detailed descriptions of the datasets, baselines and metrics are provided in Appendix I.

The results, presented in Table 2, indicate that GGFlow effectively captures the distribution of molecular data, showing significant improvements over the baselines. The high Validity without correction

suggests that GGFlow successfully learns chemical valency rules. Additionally, GGFlow achieves superior NSPDK and FCD scores on both datasets, demonstrating its ability to generate molecules with distributions closely resembling those of natural molecules. Visualizations of molecules generated by different models are shown in Figure 2, with additional results on GGFlow provided in Appendix K.

Table 2: Generation results on the QM9 and ZINC250k datasets. Results are the means of 3 different runs. The best results and the second-best results are marked **bold** and bold.

| Method | QM9 | | | | ZINC250k | | | | Step |
|---|---|---|---|---|---|---|---|---|---|
| | Val. | Val. w/o corr. | NSPDK | FCD | Val. | Val. w/o corr. | NSPDK | FCD | |
| Training Set | 100 | 100 | 0.0001 | 0.040 | 100 | 100 | 0.0001 | 0.062 | - |
| GraphAF | 100 | 67.14 | 0.0218 | 5.246 | 100 | 67.92 | 0.0432 | 16.128 | - |
| GraphDF | 100 | 83.14 | 0.0647 | 10.451 | 100 | 89.72 | 0.1737 | 33.899 | - |
| MolFlow | 100 | 92.03 | 0.0169 | 4.536 | 100 | 63.76 | 0.0468 | 20.875 | - |
| GraphEBM | 100 | 8.78 | 0.0287 | 6.402 | 100 | 5.29 | 0.2089 | 35.467 | - |
| PS-VAE | - | - | 0.0077 | 1.259 | - | - | 0.0112 | 6.320 | - |
| MolHF | - | - | - | - | 100 | 93.62 | 0.0387 | 23.940 | - |
| EDP-GNN | 100 | 47.69 | 0.0052 | 2.683 | 100 | 83.16 | 0.0483 | 16.819 | 1000 |
| GDSS | 100 | 96.17 | 0.0033 | 2.565 | 100 | 97.12 | 0.0192 | 14.032 | 1000 |
| GSDM | 100 | 99.90 | 0.0034 | 2.614 | 100 | 92.57 | 0.0168 | 12.435 | 1000 |
| GruM | 100 | 99.69 | **0.0002** | 0.108 | 100 | 98.32 | 0.0023 | 2.235 | 1000 |
| SwinGNN | 100 | 99.66 | 0.0003 | 0.118 | 100 | 86.16 | 0.0047 | 4.398 | 500 |
| DiGress | 100 | 98.29 | 0.0003 | **0.095** | 100 | 94.98 | 0.0021 | 3.482 | 500 |
| GGFlow | 100 | **99.91** | **0.0002** | 0.148 | 100 | **99.63** | **0.0010** | **1.455** | 500 |

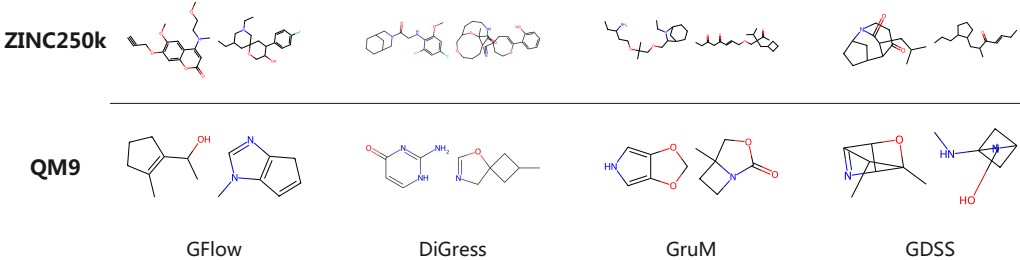

Figure 2: Visualization of generated samples of different models in different molecular datasets

## 4.3 CONDITIONAL GENERATION

To further evaluate the performance of our model, we conducted conditional generation experiments on the QM9 dataset, focusing on generating molecules with molecular properties $\mu$ that closely match a target value $\mu^*$. In the experiment, we set the target value as 1, i.e. $\mu^* = 1$.

For the experiment, we employed a reinforcement learning-based guidance method and compared it to the guided version of DiGress, which also proposes an effective approach for discrete diffusion models in conditional generation tasks. The reward function was defined as $|\mu - \mu^*|$, and the model was trained over 10,000 steps using the training settings detailed in Section 4.2. To evaluate the effectiveness of our guidance method, we compared it against three baselines: (1) Guidance for DiGress (Vignac et al., 2022). (2) Direct supervised training (ST) (3) Supervised fine-tuning (SFT). Additionally, we calculated the mean and variance of $|\mu - \mu^*|$ for samples generated unconditionally by both DiGress and GGFlow to provide a baseline comparison. Further details of the experiment are provided in Appendix I.5.

The results, detailed in Table 3, demonstrate the superiority of our reinforcement learning-based conditional generation method over both ST and SFT approaches. Notably, our method surpasses the guidance techniques used in diffusion models, showcasing its enhanced ability to steer the generative

process toward desired outcomes. Additionally, our approach achieves higher validity in conditional generated tasks, highlighting its robustness and superior performance in goal-directed generation.

Table 3: Mean absolute error of molecular property $\mu$ on conditional generation on the QM9 dataset.

| Methods | DiGress | | GGFlow | | | |
| --- | --- | --- | --- | --- | --- | --- |
| | Uncondition | +Guidance | Unconditition | Supervised Training | +SFT | +RL |
| Mean | 1.562 | 1.092 | 1.569 | 1.184 | 1.223 | **0.672** |
| Variance | 1.641 | 0.894 | 1.987 | 1.579 | 1.893 | **0.647** |
| Val. w/o corr. | 96.54 | 74.2 | 98.93 | 86.1 | 87.0 | **92.2** |

## 4.4 ABLATION STUDIES

To validate the efficiency and effectiveness of GGFlow, we conducted a series of ablation experiments using the Community-small and Planar datasets, focusing on: (1) the results of varying inference steps, (2) the model performance without the integration of Optimal Transport (OT), denoted as GGFlow (w/o OT), and (3) the model performance without the GraphEvo module, denoted as GGFlow (w/o Evo). (4) the model performance without the GraphEvo module and optimal transport, denoted as GGFlow (w/o both). The results of these studies are depicted in Figure 3 and detailed in Table 4. Additional details about the experimental settings are provided in Appendix J.2.

The results for varying inference steps are illustrated in Figure 3. Our findings indicate that GGFlow outperforms other diffusion-based models with fewer inference steps, such as 100 and 200, highlighting GGFlow's enhanced sampling efficiency. Furthermore, the integration of Optimal Transport significantly boosts sampling efficiency and enhances generative performance and stability, as evidenced by performance gains in both generic graph generation tasks. Further inference results for the Planar dataset, which reinforce the advantages of Optimal Transport, can be found in Appendix J.2. Moreover, the GraphEvo module improves the performance of

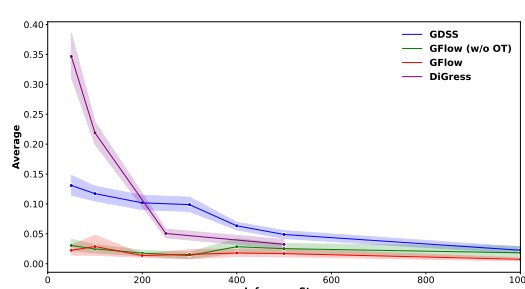

Figure 3: Ablation Studies of varying inference steps on Community-small dataset.

GGFlow. The triangle attention mechanism for edges in GraphEvo captures more complex node and edge features, leading to substantial performance improvements. Moreover, even without the combined use of GraphEvo and Optimal Transport, our methods still outperform DiGress, highlighting the advantages of flow matching over traditional diffusion models. A detailed analysis of the training procedure, which illustrates the stability provided by Optimal Transport, is also included in Appendix J.2..

Table 4: Ablation studies on the OT and GraphEvo on the Commuinty-small and Planar datasets. Results are the means of 3 different runs. The best results are marked **bold**.

| Method | Community-small | | | Planar | | | | | Step |
| --- | --- | --- | --- | --- | --- | --- | --- | --- | --- |
| | Deg. | Clus. | Orbit | Deg. | Clus. | Orbit | Spec. | Val.&Nov.&Uni. | |
| DiGress | 0.032 | 0.047 | 0.009 | **0.0003** | 0.0372 | 0.0098 | 0.0106 | 87.5 | 500 |
| GGFlow (w/o both) | 0.029 | 0.076 | 0.003 | 0.0023 | 0.1076 | 0.0053 | 0.0099 | 92.5 | 500 |
| GGFlow (w/o OT) | 0.028 | 0.027 | 0.007 | **0.0015** | 0.0431 | 0.0020 | **0.0067** | 97.0 | 500 |
| GGFlow (w/o Evo) | 0.018 | 0.075 | 0.004 | 0.0020 | 0.0763 | 0.0034 | 0.0124 | 94.5 | 500 |
| GGFlow | **0.001** | 0.084 | **0.004** | 0.0156 | **0.0196** | **0.0019** | 0.0091 | **97.5** | 500 |

## 5 CONCLUSION

In this paper, we introduced GGFlow, a discrete flow matching generative model for graphs that incorporates optimal transport and an innovative graph transformer network. GGFlow achieves state-of-the-art performance in unconditional graph generation tasks. Additionally, we presented a novel guidance method using reinforcement learning to control the generative trajectory toward a preferred distribution. Furthermore, our model demonstrates the ability to achieve the best performance across various tasks with fewer inference steps compared to other baselines which highlights the practical impact of our guidance method. A primary limitation is scalability to larger graphs ($|\mathcal{V}| > 500$), attributable to the increased time complexity from triangle attention updates and spectral feature computations. Generation times for different graph scales are provided in Appendix J.3. Future work will focus on enhancing our model's scalability in larger graphs.

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

APPENDIX

## A  BACKGROUND

### A.1  CONTINUOUS FLOW MATCHING GENERATIVE MODEL

The generative model aims to establish a mapping $f : \mathbb{R}^d \to \mathbb{R}^d$ that transforms a noise distribution $q_0$ into a target data distribution $q_1$. This transformation is dependent on a density function $p_0$ over $\mathbb{R}^d$, and an integration map $\psi_t$, which induces a pushforward transformation $p_t = [\psi_t]_\# (p_0)$. This denotes the density of points $x \sim p_0$ transported from time 0 to time $t$ along a vector field $u : [0, 1] \times \mathbb{R}^d \to \mathbb{R}^d$.

The vector field $u$ is formulated as:
$$\mathrm{d}x = u_t(x)\mathrm{d}t.$$

The solution $\psi_t(x)$ to this ODE, with the initial condition $\psi_0(x) = x$, represents the trajectory of the point $x$ governed by $u$ from time 0 to time $t$.

The evolution of the density $p_t$, viewed as a function $p : [0, 1] \times \mathbb{R}^d \to \mathbb{R}$, is encapsulated by the continuity equation:
$$\frac{\partial p}{\partial t} = -\nabla \cdot (p_t u_t),$$

with the initial condition given by $p_0$. Here, $u$ is the probability flow ODE for the path of marginal probabilities $p$, generated over time.

In practical applications, if the probability path $p_t(x)$ and the generating vector field $u_t(x)$ are known and $p_t(x)$ is tractably sampled, we leverage a time-dependent neural network $v_\theta(\cdot, \cdot) : [0, 1] \times \mathbb{R}^d \to \mathbb{R}^d$ to approximate $u$. The neural network is trained using the flow matching objective:
$$\mathcal{L}_{\mathrm{FM}}(\theta) = \mathbb{E}_{t \sim \mathcal{U}(0,1), x \sim p_t(x)} \|v_\theta(t, x) - u_t(x)\|^2, \tag{11}$$

which enhances the model's capability to simulate the target dynamics accurately. Avoiding the explicit construction of the intractable vector field, recent works express the probability path as a marginal over a joint involving a latent variable $z$: $p(x_t) = \int p(z)p_{t|z}(x_t|z)$. (Lipman et al., 2022; Tong et al., 2023) and the $p_{t|z}(x_t|z)$ is a conditional probability path, satisfying some boundary conditions at $t = 0$ and $t = 1$.

The conditional probability path also satisfies the transport equation with the conditional vector field $u_t(x|x_1)$:
$$\frac{\partial p_t(x|x_t)}{\partial t} = -\nabla \cdot (u_t(x|x_1)p_t(x_t|x_1)). \tag{12}$$

We can construct the marginal vector field $u_t(x)$ via the conditional probability path $p_{t|1}(x_t|x_1)$ as:
$$u_t(x) = \mathbb{E}_{x_1 \sim p_{1|t}}[u_t(x|x_1)]. \tag{13}$$

We can replace the flow matching loss $\mathcal{L}_{\mathrm{FM}}$ with an equivalent loss regressing the conditional vector field $u_t(x|x_1)$ and marginalizing $x_1$ instead:
$$\mathcal{L}_{\mathrm{CFM}}(\theta) = \mathbb{E}_{\mathcal{U}(t;0,1), x_1 \sim q, x_t \sim p_t(x|x_1)}[u_\theta(t, x) - u_t(x|x_1)].$$
$$\nabla_\theta \mathcal{L}_{\mathrm{FM}}(\theta) = \nabla_\theta \mathcal{L}_{\mathrm{CFM}}(\theta).$$

So we can use $\mathcal{L}_{\mathrm{CFM}}(\theta)$ instead to train the parametric vector field $u_\theta$.

## B  RELATED WORKS

### B.1  COMPARISON WITH DISCRETE FLOW MATCHING

Campbell et al. (2024) first introduced flow matching in discrete spaces using a continuous-time Markov chain. Building on this, Gat et al. (2024) expanded the framework to encompass general

source and target couplings, including U-coupling and C-coupling. GGFlow advances discrete flow matching and its source-target coupling to suit graph-structured data. Our approach innovatively incorporates efficient optimal transport for graphs within the flow matching framework. To address the inherent sparsity and permutation invariance of graphs, we employ a tailored prior distribution. Additionally, we implement an edge-augmented graph transformer to enhance generative performance and adopt a goal-guided framework for conditional generation. These advancements collectively enhance the practical applicability of GGFlow.

### B.2 COMPARISON WITH GRAPH DISCRETE DIFFUSION MODEL

DiGress (Vignac et al., 2022) and PPGN (Haefeli et al., 2022) were among the first to apply discrete diffusion models to graph generation, highlighting the advantages of discrete state spaces. DiGress further introduced an optimal prior distribution and global structural features specifically designed to enhance graph generation. Their forward and generative processes are expressed as:

$$q(G^t|G^0) = \text{Cat}(G^t, p = G^0 \bar{Q}_t), \text{ with } \bar{Q}_t = Q_1 Q_2 \ldots Q_t, \tag{14}$$

$$q(G^{t-1}|G^t, G^0) = \frac{q(G^t|G^{t-1}, G^0)q(G^{t-1}|G^0)}{q(G^t|G^0)} = \text{Cat}(G^{t-1}; p = \frac{G^t Q_t^T \odot G^0 \bar{Q}_{t-1}}{G^0 \bar{Q}_t G^{t*}}), \tag{15}$$

where $G^t$ represents the noisy graph at time $t$, and $Q_t$ is the time-dependent transition matrix. These methods require maintaining convergence properties of transition matrix and cumulative matrix products, constraining the choice of prior distributions and destabilizing training. In contrast, GGFlow employs a simpler interpolation between the prior and data distributions during training, avoiding cumulative products and improving both training stability and the ease of selecting appropriate priors.

### B.3 COMPARISON WITH GRAPH DISCRETE FLOW MODEL

GraphDF (Luo et al., 2021) uses a discrete flow model to generate molecular graphs by sequentially sampling discrete latent variables and mapping them to nodes and edges via invertible modulo-shift transforms. GGFlow simplifies this by transforming the invertible modulo-shift into a conditional vector field that interpolates between the prior and data distributions, bypassing the need for complex invertible mappings. Furthermore, while GraphDF adopts an autoregressive process for graph generation, GGFlow generates the entire graph in a one-shot manner, capturing holistic relationships among nodes and edges more efficiently.

### B.4 COMPARISON WITH GRAPH VARIATIONAL FLOW MATCHING

CatFlow (Eijkelboom et al., 2024) employs variational inference to apply flow matching to categorical data, but it only considers the conditional vector field under the assumption of independent coupling in the joint distribution $\pi(G^0, G^1)$ and fails to consider the inherent sparsity of graph structures. GGFlow extends this by generalizing $\pi(G^0, G^1)$ as a 2-Wasserstein optimal transport map and incorporating an optimal prior distribution tailored for graph structures, improving performance in generation tasks. Additionally, GGFlow introduces a novel goal-directed approach for discrete flow matching in conditional generation tasks, enhancing its practical applicability.

## C PROOFS

### C.1 OPTIMAL PRIOR DISTRIBUTION

This prior is structured as a product of a single distribution $v$ for all nodes and a single distribution $e$ for all edges, $\prod_i v \times \prod_{i,j} e$, to ensure exchangeability across the graph components.

**Theorem 2** (Optimal prior distribution). *Consider the class $\mathcal{C} = \{\prod_i u \times \prod_{i,j} v, (u,v) \in \mathcal{P}(\mathcal{V}) \times \mathcal{P}(\mathcal{E})\}$ of distributions over graphs, which factorize as the product of a uniform distribution $v$ over node attribute space $\mathcal{V}$ and a uniform distribution $e$ over edge attribute space $\mathcal{E}$. Given any arbitrary distribution $P$ over graphs (viewed as a tensor of order $n + n^2$), with $q_V$ and $q_E$ as its marginal distributions for node and edge attributes respectively, then the orthogonal projection of $P$ onto $\mathcal{C}$ is*

*defined as $\phi^G = \prod_i q_V \times \prod_{i,j} q_E$. This projection minimizes the Euclidean distance:*

$$\phi^G \in \arg\min_{(v,e)\in\mathcal{C}} \|P - \prod_{1\leq i\leq n} v \times \prod_{1\leq i,j\leq n} e\|_2^2.$$

The details and proof of Theorem 2 are extensively discussed in DiGress (Vignac et al., 2022).

## C.2 CHOICE OF CONDITIONAL VELOCITY FIELD

In GGFlow, the conditional vector field for discrete flow matching is defined as (Campbell et al., 2024):

$$u_t(G, G^t|G^0, G^1) = \frac{\text{ReLU}(\partial_t p_{t|1}(G|G^1) - \partial_t p_{t|1}(G^t|G^1))}{\mathbf{Z}_t \cdot p_{t|1}(G^t|G^1)}$$

$$= \frac{1}{\mathbf{Z}_t(1-t)p_{\text{ref}}}\delta\{G, G^1\}(1 - \delta\{G^t, G^1\}), G_t \neq G,$$

where $\text{ReLU}(a) = \max(a, 0)$ and $\mathbf{Z}_t = |\{G^t : p_t(G^t|G^0, G^1) > 0\}|$. $u_t(G, G^t|G^0, G^1) = 0$ when $p_t(G|G^1, G^0) = 0$ and $p_t(G^t|G^1, G^0) = 0$. When $G^t = G$, the rate matrix $R(G^t, G^t|G^0, G^1) = -\sum_{G^t\neq G} R(G^t, G|G^0, G^1)$. For simplification, the graph $G$ is denoted as variable $x$

*Proof.* Consider the conditional probability $p_{t|1}(x^t|x^1, x^0) = p_t(x^t|x^1, x^0) = \text{Cat}\Big(t\delta\{x^1, x^t\} + (1-t)q_x\Big)$, where $q_x$ is the prior distribution. We derive its time derivative:

$$\partial_t p_{t|1}(x^t|x^1, x^0) = \delta\{x^1, x^t\} - q_x, \tag{16}$$

We then construct the conditional rate matrix $u_t(x^t, x|x^1, x^0)$ as:

$$u_t(x^t, x|x^1, x^0) = \frac{\text{ReLU}(\partial_t p_{t|1}(x|x^1, x^0) - \partial_t p_{t|1}(x^t|x^1, x^0))}{\mathbf{Z}_t \cdot p_{t|1}(x^t|x^1, x^0)}$$

$$= \frac{\text{ReLU}(\delta\{x, x^1\} - q_x - \delta\{x^t, x^1\} + q_x)}{\mathbf{Z}_t(t\delta\{x^1, x^t\} + (1-t)q_x)}$$

$$= \frac{\text{ReLU}(\delta\{x, x^1\} - \delta\{x^t, x^1\})}{\mathbf{Z}_t(t\delta\{x^1, x^t\} + (1-t)q_x)}.$$

The expression simplifies under the assumption that $x^t \neq x$. The only non-zero values occur when $x = x^1$ and $x^t \neq x^1$, thus yielding:

$$u_t(x^t, x|x^1) = \frac{1}{\mathbf{Z}_t(1-t)q_x}\delta\{x, x^1\}(1 - \delta\{x^t, x^1\}), x_t \neq j \tag{17}$$

where $\mathbf{Z}_t = |\{x^t : p_t(x^t|x^1, x^0) > 0\}|$. □

## C.3 PROOF OF PROPOSITION 1

*Proof.* The Kolmogorov forward equations for discrete flow matching are expressed as:

$$\partial_t p_t = u_t p_t, \tag{18}$$

If we establish the permutation invariance of the prior distributions $p_{\text{ref}}$ and the permutation equivariance of conditional flow probabilities, then it follows that $p(G^1)$ is permutation exchangeable.

According to the Theorem 2, we deduce the permutation invariance of the prior distribution $p_{\text{ref}}$. Given the conditional probabilities $p(G^{t+\Delta t}|G^t) = \text{Cat}\Big(\delta\{G^t, G^{t+\Delta t}\} + \hat{u}_t(G^t, G^{t+\Delta t})\Delta t\Big)$, it

suffices to demonstrate the permutation equivariance of the conditional probabilities. This requires showing the permutation equivariance of the vector field $u_t$. Consider the case for nodes:

$$\pi u_t^V(V_i^t, V_i^{t+\Delta t}) = \pi \left( \mathbb{E}_{\hat{p}_{1|t}^V(V_i^1|V_i^t)}[u_t^V(V_i^t, V_i^{t+\Delta t}|V_i^1, V_i^0)] \right),$$

$$\text{LHS} = u_t^V(V_{\pi^{-1}(i)}^t, V_{\pi^{-1}(i)}^{t+\Delta t}),$$

$$\text{RHS} = \left( \mathbb{E}_{\hat{p}_{1|t}^V(V_{\pi^{-1}(i)}^1|V_{\pi^{-1}(i)}^t)}[u_t^V(V_{\pi^{-1}(i)}^t, V_{\pi^{-1}(i)}^{t+\Delta t}|V_{\pi^{-1}(i)}^1, V_{\pi^{-1}(i)}^0)] \right),$$

$$= u_t^V(V_{\pi^{-1}(i)}^t, V_{\pi^{-1}(i)}^{t+\Delta t}) = \text{LHS}.$$

where $\pi$ is a permutation operator. This establishes the permutation equivariance of $u_t$ and the exchangeability of the generated distribution. $\square$

### C.4 PROOF OF THEOREM 1

First, we want to clarify the rationale and foundation of our theorem. The goal of optimal transport is to pair source and target data points with minimal cost during training, which is beneficial for our interpolation (Bose et al., 2023; Song et al., 2024). Thus, we design our optimal transport approach from the perspective of interpolation.

We define the node order of the graph $G$ as the order of the nodes and edges in matrix representation. For example, if the node set of $G$ is $\{A, B, C\}$, the possible node orders include $(A, B, C)$, $(B, A, C)$ or $(C, B, A)$.

In the interpolation process, we transform the graph representation to a matrix representation before performing interpolation. For example, for source data $G^0 = (V^0, E^0), V^0 \in \mathbb{R}^{a \times n}, E^0 \in \mathbb{R}^{a \times a \times m}$ and target data $G^1 = (V^1, E^1), V^1 \in \mathbb{R}^{a \times n}, E^1 \in \mathbb{R}^{a \times a \times m}$, where $a$ is the number of nodes, $n$ is the class number of nodes, and $m$ is the class number of edges, the node orders of $G^0$ and $G^1$ have been fixed. Therefore, interpolation is performed directly on these fixed node orders.

The optimal transport aims to find pairs with the minimum cost for interpolation, and the interpolation is conducted on a fixed node order. Additionally, during optimal transport calculations, we also utilize the matrix representation of these graphs and our prior distribution is permutation invariant. Therefore, we aim to match source data with the target data $G^1$ whose node order is fixed, to achieve minimal transport cost. Furthermore, we assume that all pairs of source and target data share the same node order during optimal transport, which also facilitates the identification of pairs with minimal cost.

Regarding the permutation of the intermediate graph $G^t$, we have $\pi G^t = t\pi G^0 + (1-t)\pi G^1$, where $G^0$ and $G^1$ share an identical permutation. Our network $\hat{p}^1(G^1|G^0, G^t)$ needs to maintain permutation equivariance, such that $\hat{p}^1(G^1|\pi G^0, \pi G^t) = \pi \hat{p}^1(G^1|G^0, G^t)$ for any permutation $\pi$ to approximate $\pi G^1$. So we prove the invariance of optimal transport under identical permutations, i.e. $\phi(G^0, G^1) = \phi(\pi G^0, \pi G^1)$.

*Proof.* Building on the foundations established in Theorem 2 and Proposition 1, we confirm the permutation invariance of both the target and source distributions. The Hamming distance is invariance under identical permutations $\pi$, as shown by:

$$H(G^0, G^1) = \sum_i \delta(v_i^0, v_i^1) + \frac{1}{2} \sum_{i,j} \delta(e_{ij}^0, e_{ij}^1)$$

$$= \sum_i \delta(v_{\pi^{-1}(i)}^0, v_{\pi^{-1}(i)}^1) + \frac{1}{2} \sum_{i,j} \delta(e_{\pi^{-1}(i)\pi^{-1}(j)}^0, e_{\pi^{-1}(i)\pi^{-1}(j)}^1)$$

$$= H(\pi G^0, \pi G^1)$$

This property of the Hamming distance ensures the invariance of the optimal transport map $\phi^*$ under identical permutations.

$\square$

Additionally, the prior distribution is permutation invariant and our GraphEvo is permutation equivariance, all permutations of graphs are generated with equal probability (Eijkelboom et al., 2024).

**Lemma 1.** *Let $p_0(G)$ be an exchangeable distribution and our model $\hat{p}_{1|t}(G^1|G^t, G^0)$ is permutation equivariant. Then, all permutations of generated graphs with equal probability.*

*Proof.* As the permutation equivariance of our model $\hat{p}_{1|t}(G^1|G^t, G^0)$, implies the equivariance of our vector fields $u_t$. Moreover, the sampling procedure exhibits permutation equivariance, where $\pi$ is a permutation.

$$\hat{p}_{1|t}(G^1|\pi G^0) = \pi\hat{p}_{1|t}(G^1|G^0), \quad t = 0$$

$$p_{\Delta t|0}(G^{\Delta t}|\pi\hat{G}^1, \pi G^0) = \delta\{\cdot, \pi G^0\} + u_0(\cdot, \pi G^0|\pi G^0, \pi\hat{G}^1)\Delta t$$

$$= \pi[\delta\{\cdot, G^0\} + u_0(\cdot, G^0|G^0, \hat{G}^1)\Delta t] = \pi p_{\Delta t|0}(G^{\Delta t}|\hat{G}^1, G^0), \quad t = 0$$

$$p_{t+\Delta t|t}(G^{\Delta t+t}|\pi\hat{G}^1, \pi G^t, \pi G^0) = \delta\{\cdot, \pi G^t\} + u_t(\cdot, \pi G^t|\pi G^0, \pi\hat{G}^1)\Delta t$$

$$= \pi[\delta\{\cdot, G^t\} + u_t(\cdot, G^t|G^0, \hat{G}^1)\Delta t] = \pi p_{t+\Delta t|t}(G^{\Delta t+t}|\hat{G}^1, G^t, G^0), \quad t = \Delta t, \dots, 1 - \Delta t$$

Therefore, since $p_0$ assigns equal density of all permutations of $G$, the resulting distribution $p_1$ preservers this property. □

## D DETAILS OF GRAPHEVO

GraphEvo is a novel edge-augmented graph transformer model designed for graph data. To enhance the generative capabilities of GGFlow, GraphEvo introduces a triangle update mechanism, which significantly improves the exchange of edge information. We incorporate FiLM and PNA layers into our architecture (Vignac et al., 2022):

$$\text{FiLM}(X_1, X_2) = X_1(\text{Linear}(X_2) + 1) + \text{Linear}'(X_2)$$

$$\text{PNA}(X) = \text{Linear}\Big(\text{Cat}(\max(X), \min(X), \text{mean}(X), \text{std}(X))\Big).$$

The full architecture of GraphEvo is illustrated in Algorithm 3 and is permutation equivariant. The time complexity of GraphEvo is $O(N^3)$.

---

**Algorithm 3** Architecture of GraphEvo

---

**Require:** $G, t, N_{\text{layer}}$
1: $\mathbf{V}, \mathbf{E} \leftarrow G$
2: $\mathbf{y} \leftarrow \text{ExtractFeature}(G), \mathbf{t} \leftarrow \text{TimeEmbedding}(t)$
3: $\mathbf{y} \leftarrow \mathbf{y} + \mathbf{t}$
4: $\mathbf{X}, \mathbf{E}, \mathbf{y} \leftarrow \text{Linear}(V), \text{Linear}(\mathbf{E}), \text{Linear}(\mathbf{y})$
5: **for** $t = 0, 1, \dots, N_{\text{layer}}$ **do**
6: $\quad \mathbf{X}', \mathbf{E}', \mathbf{y}' \leftarrow \text{SelfAttention}(\mathbf{X}, \mathbf{E}, \mathbf{y})$
7: $\quad \mathbf{X} \leftarrow \text{ReLU}\Big(\text{LayerNorm}(\mathbf{X} + \text{Dropout}(\mathbf{X}'))\Big)$
8: $\quad \mathbf{E} \leftarrow \text{ReLU}\Big(\text{LayerNorm}(\mathbf{E} + \text{Dropout}(\mathbf{E}'))\Big)$
9: $\quad \mathbf{y} \leftarrow \text{ReLU}\Big(\text{LayerNorm}(\mathbf{y} + \text{Dropout}(\mathbf{y}'))\Big)$
10: **end for**
11: $\hat{p}_{1|t}^V(V^1|V^t, V^0), \hat{p}_{1|t}^E(E^1|E^t, E^0), \mathbf{y} \leftarrow \text{Linear}(V), \text{Linear}(\mathbf{E}), \text{Linear}(\mathbf{y})$
12: **return** $\hat{p}_{1|t}^V(V^1|V^t, V^0), \hat{p}_{1|t}^E(E^1|E^t, E^0), \mathbf{y}$

---

GraphEvo integrates global structural features to improve generation performance, including both graph-theoretic and domain-specific attributes:

**Graph-theoretic features**: These encompass node-level properties such as the number of $k$-cycles ($k \leq 5$) containing this point and an estimate of the largest connected component, alongside graph-level metrics like the total number of $k$-cycles ($k \leq 6$) and connected components.

**Molecular features**: These account for the current valency of each atom and the molecular weight of the entire molecule.

### D.1  PROOF OF PROPOSITION 2

*Proof.* Let $G^t = (V^t, E^t)$ is a intermediate graph, and $\pi G^t = (\pi^*V, \pi^*E\pi)$ is the permutation. To prove the permutation properties of the graph, we need to consider two aspects: additional structural features and the model architecture.

First, the spectral and structural features are permutation equivariant for node-level features and invariant for graph-level features. Additionally, the FiLM blocks and Linear layers are permutation equivariant, while the PNA pooling function is permutation invariant. Layer normalization is also permutation equivariant.

As GraphEvo is built using permutation equivariant components, we conclude that the overall model is permutation equivariant.

□

### E  TIME COMPLEXITY OF OPTIMAL TRANSPORT

To analyze the time complexity of optimal transport (OT), we compared the training time of OT with that of DiGress, using identical architectures on an NVIDIA A100 80G GPU. We evaluated the effects of model size, batch size, and number of nodes by measuring the duration of single training steps across three different datasets. Our results indicate that the time required for OT accounts for only 5% of the total training time, highlighting the efficiency of our optimal transport.

Table S1: Time Complexity of Optimal Transport

| Dataset | Planar | Zinc250k | Community-small |
|---|---|---|---|
| DiGress Training Time (s) | 0.1647 | 0.1690 | 0.0456 |
| GGFlow Training Time (s) | 0.1264 | 0.1301 | 0.0408 |
| Optimal Transport Time (s) | 0.0025 | 0.0070 | 0.0024 |
| Percentage of OT | 1.9% | 5.3% | 5.6% |
| Model Size (M) | 3.6 | 4.6 | 6.4 |
| Batch Size | 64 | 128 | 80 |
| Number of Nodes | 64 | [6,38] | [12,20] |

### F  SOURCE CODE

The code will be made publicly available upon the publication of this paper.

### G  TOY EXAMPLE OF GOAL-GUIDED GRAPH GENERATION

We demonstrate the utility of our goal-guided framework of flow matching with a toy example, depicted in Figure S1: (a) shows a trained unconditional flow matching model mapping noise distribution $p_0$ to data distribution $p_1$. (b, c) illustrate the effect of temperature $T$ on the exploration, with higher temperatures resulting in broader data point distribution. (d) shows how fine-tuning according to Equation 10 concentrates data in regions with higher rewards. (e-f) illustrate the corresponding flow matching trajectories.

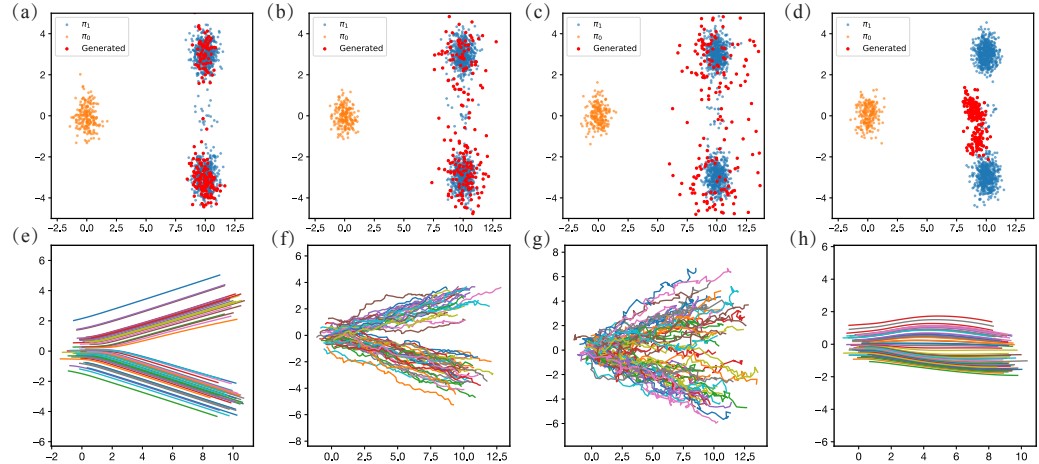

Figure S1: (a-d) Data distribution of the flow matching model, $\pi_0$ is the original distribution (orange), $\pi_1$ is the target data distribution (blue), and the red dots are the data distribution generated by the model. (e-h) In reinforcement learning, the flow matching model conducts exploration/sampling trajectories

## H  ADDITIONAL EXPERIMENTS RESULTS

In this section, we present additional metrics including Spectre (Spec.)  and Validity&Novelty&Uniquess (Val.&Nov.&Uni.) across general graph datasets including the Planar and Enzymes datasets, as summarized in Tables S2, S3 and S4. The MMD kernel in the planar dataset followed the GruM (Jo et al., 2024). We also include the standard deviation of our results in Table S5, illustrating the consistency and superior performance of our method.

To further compare GGFlow with baseline models, we measured the MMD between the test datasets and a set of 1,024 generated graphs in the Ego-small and Community-small datasets. The results in Table S6 demonstrate that GGFlow achieves the highest performance across all metrics, significantly outperforming other baseline models.

Table S2: Additional generation results on the generic graph datasets. Results are the means of 3 different runs. The best results are marked **bold**.

| Method | Ego-small | | Community-small | | Grid | | Step |
|---|---|---|---|---|---|---|---|
| | Spec. | Nov.&Uni. | Spec. | Nov.&Uni. | Spec. | Nov.&Uni. | |
| Training Set | 0.006 | 100 | 0.012 | 100 | 0.009 | 25 | - |
| GDSS | 0.034 | 27.5 | 0.053 | 100.0 | 0.043 | 100.0 | 1000 |
| GSDM | - | - | **0.024** | 0.0 | 0.015 | 0.0 | 1000 |
| DiGress | 0.017 | **100.0** | 0.055 | 100.0 | 0.025 | 100.0 | 500 |
| SwinGNN | 0.016 | 52.5 | 0.025 | 55.0 | **0.008** | 100.0 | 500 |
| GGFlow | **0.006** | 32.5 | 0.031 | **100.0** | 0.022 | **100.0** | 500 |

## I  IMPLEMENT DETAILS

### I.1  ALGORITHMS OF GGFLOW

Details of the training procedure and guided training procedure are provided in Algorithm 4 and 5.

Table S3: Generation results on the planar graph datasets. The best results are marked **bold**. $\sigma$ denotes the standard deviation.

| Method | Planar | | | | | Step |
|--------|--------|--------|--------|--------|--------------|------|
| | Deg. | Clus. | Orbit | Spec. | Val.&Nov.&Uni. | |
| Training Set | 0.0002 | 0.0165 | 0.0002 | 0.0050 | 100 | - |
| GDSS | 0.0039 | 0.2593 | 0.1732 | 0.0370 | 0.0 | 1000 |
| GRASP | 0.0022 | 0.2749 | 0.0055 | 0.0098 | 0.0 | 200 |
| DiGress | **0.0003** | 0.0372 | 0.0098 | 0.0106 | 87.5 | 500 |
| GruM | 0.0004 | 0.0382 | 0.0095 | **0.0069** | 75.0 | 1000 |
| GGFlow | 0.0156 | **0.0196** | **0.0019** | 0.0091 | **97.5** | 500 |
| $\sigma$ | 0.0064 | 0.0037 | 0.0006 | 0.0012 | 2.5 | - |

Table S4: Generation results on the Enzymes graph datasets. The best results are marked **bold**. $\sigma$ denotes the standard deviation.

| Method | Enzymes | | | | Step |
|--------|-------|-------|-------|-------|------|
| | Deg. | Clus. | Orbit | Avg. | |
| Training Set | 0.008 | 0.096 | 0.012 | 0.039 | - |
| GraphRNN | 0.017 | 0.043 | 0.021 | 0.043 | - |
| GraphAF | 1.669 | 1.283 | 0.266 | 1.073 | - |
| GraphDF | 1.503 | 1.061 | 0.202 | 0.922 | - |
| GraphVAE | 1.369 | 0.629 | 0.191 | 0.730 | - |
| EDP-GNN | 0.023 | 0.268 | 0.082 | 0.124 | 1000 |
| GDSS | 0.026 | 0.102 | 0.009 | 0.046 | 1000 |
| GSDM | 0.013 | 0.088 | 0.010 | 0.037 | 1000 |
| DiGress | 0.010 | 0.046 | 0.002 | 0.019 | 500 |
| GGFlow | **0.008** | **0.026** | **0.002** | **0.012** | 500 |
| $\sigma$ | 0.0041 | 0.0106 | 0.0008 | 0.0130 | - |

Table S5: Standard deviation and mean of generation results on the general graph datasets. $\mu$ and $\sigma$ denote the mean and standard deviation, respectively

| Metric | Ego-small | | | | Community-small | | | | Grid | | | |
|--------|------|-------|-------|-------|------|-------|-------|-------|------|-------|-------|-------|
| | Deg. | Clus. | Orbit | Sepc. | Deg. | Clus. | Orbit | Sepc. | Deg. | Clus. | Orbit | Sepc. |
| $\mu$ | 0.005 | 0.033 | 0.005 | 0.008 | 0.011 | 0.030 | 0.002 | 0.031 | 0.030 | 0.000 | 0.016 | 0.022 |
| $\sigma$ | 0.007 | 0.012 | 0.003 | 0.001 | 0.006 | 0.012 | 0.002 | 0.002 | 0.008 | 0.000 | 0.003 | 0.001 |

## I.2    BASELINES IMPLEMENTATION

To benchmark the performance of GGFlow, we ensure consistency by using identical splits of training and test sets across all datasets. Below, we provide the implementation details for each baseline model. To guarantee a fair comparison, most baseline models are retrained three times, and the average results from these runs are reported as the final outcomes in unconditional generation tasks. The results of the DeepGMG, GraphRNN and GNF for Ego-small and Community-small dataset are taken from their original papers.

**GraphAF** (Shi et al., 2019)   We follow the implementation guidelines provided in the TorchDrug tutorials (`https://torchdrug.ai/docs/tutorials/generation.html`).

**GraphDF** (Shi et al., 2019)   Model scripts are sourced from the DiG repository (`https://github.com/divelab/DIG/tree/dig-stable/examples/ggraph/GraphDF`).

Table S6: Generation results on the generic graph datasets with 1024 generated graphs. The best results are marked **bold**.

| Method | Ego-small | | | | Community-small | | | | Step |
|--------|------|------|------|------|------|------|------|------|------|
| | Deg. | Clus. | Orbit | Sepc. | Deg. | Clus. | Orbit | Spec. | |
| GraphRNN | 0.040 | 0.050 | 0.060 | - | 0.030 | 0.010 | 0.010 | - | - |
| GNF | 0.010 | 0.030 | 0.001 | - | 0.120 | 0.150 | 0.200 | - | - |
| EDP-GNN | 0.010 | 0.025 | 0.003 | - | 0.006 | 0.127 | 0.018 | - | 1000 |
| GDSS | 0.023 | 0.020 | 0.005 | 0.047 | 0.029 | 0.068 | 0.004 | 0.151 | 1000 |
| GSDM | - | - | - | - | **0.003** | 0.008 | 0.0009 | 0.011 | 1000 |
| DiGress | 0.017 | 0.038 | 0.006 | 0.021 | 0.013 | 0.040 | 0.004 | 0.055 | 500 |
| SwinGNN | 0.004 | 0.023 | 0.003 | 0.023 | 0.003 | 0.088 | 0.010 | 0.016 | 500 |
| GGFlow | **0.004** | **0.004** | **0.0008** | **0.009** | 0.004 | **0.003** | **0.0006** | 0.018 | 500 |

---

**Algorithm 4** Training Procedure of GGFlow

---

**Require:** $G = (V, E), q_V, q_E,$
1: **for** $n \in \{0, \dots, N_{\text{iter}} - 1\}$ **do**
2:     $t \in \mathcal{U}(0, 1), G^1 = G$
3:     $G^0 = (V^0, E^0) \sim p^{\text{ref}}$
4:     $(G^0, G^1) \sim \text{OptimalTransport}(G^0, G^1)$
5:     // Sample from conditional probability flow.
6:     $V^t = (t\delta\{V^1, \cdot\} + (1-t)V^0)$ and $E^t = (t\delta\{E^1, \cdot\} + (1-t)E^0)$
7:     $\hat{p}_{1|t}^V(V^1|V^t, V^0), \hat{p}_{1|t}^E(E^1|E^t, E^0), \mathbf{y} = \text{GraphEvo}_{\theta_n}(V^t, E^t, t, f^t)$
8:     $\mathcal{L} = \mathbb{E}_{p_{\text{data}}(G^1)\mathcal{U}(t;0,1)\pi(G^0,G^1)p_t(G^t|G^0,G^1)}[\log \hat{p}_{1|t}(G^1|G^t, G^0)]$
9:     $\theta_{n+1} = \text{optimizer\_update}(\theta_n, \mathcal{L})$
10: **end for**
11: $\theta^* = \theta_{N_{\text{iter}}}$
12: **return** $\theta^*$

---

**GraphVAE** (Shi et al., 2019) Scripts are obtained from the GraphVAE section of the GraphRNN repository (https://github.com/JiaxuanYou/graph-generation/tree/master/baselines/graphvae).

**MoFlow** (Zang and Wang, 2020) Implementation scripts are taken from the MoFlow repository (https://github.com/calvin-zcx/moflow).

**GraphEBM** (Liu et al., 2021) We use the implementation available in the GraphEBM repository (https://github.com/biomed-AI/GraphEBM).

**EDP-GNN** (Niu et al., 2020) The model is implemented according to the scripts in the EDP-GNN repository (https://github.com/ermongroup/GraphScoreMatching).

**GDSS** (Jo et al., 2022b) Implementation details are sourced from the GDSS repository (https://github.com/harryjo97/GDSS).

**GSDM** (Luo et al., 2023) Scripts are implemented from the GSDM repository (https://github.com/ltz0120/Fast_Graph_Generation_via_Spectral_Diffusion).

**PS-VAE** (Kong et al., 2022) Implementation details are sourced from the PS-VAE repository (https://github.com/THUNLP-MT/PS-VAE).

**MolHF** (Zhu et al., 2023) The model is implemented according to the scripts in the MolHF repository (https://github.com/violet-sto/MolHF).

**GRASP** (Minello et al., 2024) Implementation details are sourced from the GRASP repository (https://github.com/lcosmo/GRASP).

**SwinGNN** (Yan et al., 2023) Implementation details are sourced from the SwinGNN repository (https://github.com/DSL-Lab/SwinGNN). The authors employ the 'gaussian_tv' MMD

---

**Algorithm 5** Training Procedure of Guided GGFlow by Reinforcement Learning

---

**Require:** $\theta_0, \theta, \alpha, \beta, T, N_{\text{steps}}, \text{traj}, G^0 \sim p_{\text{ref}}, u_t(G^t, G|G^1, G^0), T, N_{\text{train}}$
1: $\theta \leftarrow \theta_0$
2: **for** $i \in \{1, \ldots, N_{\text{train}}\}$ **do**
3: $\quad \Delta t = 1/N_{\text{steps}}$
4: $\quad$ Collect flow trajectory $\left(G^0, t = 0, \mathcal{R}(G^0)\right)$.
5: $\quad$ **for** $n \in \{0, \ldots, N_{\text{steps}} - 1\}$ **do**
6: $\quad\quad \hat{p}_{1|t}^V(V^1|V^t, V^0), \hat{p}_{1|t}^E(E^1|E^t, E^0), \mathbf{y} = \text{GraphEvo}(V^t, E^t, t)$
7: $\quad\quad$ Get $G^{t+\Delta t}$ by sampling from Equation 9.
8: $\quad\quad (V^{t+\Delta t}, E^{t+\Delta t}) = G^{t+\Delta t}$
9: $\quad\quad t = t + \Delta t$
10: $\quad\quad$ Compute the reward function $\mathcal{R}(G^{t+\Delta t})$.
11: $\quad\quad$ Collect flow trajectory $\left(G^{t+\Delta t}, t + \Delta t, \mathcal{R}(G^{t+\Delta t})\right)$.
12: $\quad$ **end for**
13: $\quad$ Update network using Equation 10.
14: $\quad t = 0$
15: **end for**
16: **return** Guided flow matching model $\theta^*$

---

kernel, whereas other methods use 'gaussian_emd' or 'gaussian'. To ensure a fair comparison, we adopt the same kernel.

**GruM** (Jo et al., 2024)  Scripts are implemented from the GruM repository (`https://github.com/harryjo97/GruM/`).

**DiGress** (Vignac et al., 2022)  The implementation is based on the DiGress repository (`https://github.com/cvignac/DiGress`).

### I.3 DETAILS OF GENERIC DATASETS

#### I.3.1 DATASET

**Ego-small**  This dataset consists of 200 small one-hop ego graphs derived from the Citeseer network (Sen et al., 2008). Each graph contains between 4 and 18 nodes.

**Community-small**  This dataset includes 100 random community graphs, each formed by two communities of equal size generated using the E-R model (Erdős et al., 1960) with a probability parameter of $p = 0.7$. The graphs range in size from 12 to 20 nodes.

**Enzymes**  The dataset comprises 587 protein graphs, with each graph representing the tertiary structure of enzymes sourced from the BRENDA database (Schomburg et al., 2004), which have between 10 and 125 nodes.

**Grid**  The dataset consists of 100 standard 2D grid graphs with $100 \leq |V| \leq 400$.

**Planar**  The dataset consists of 200 planar graphs, each with 64 nodes, generated using Delaunay triangulation on uniformly distributed random points.

Table S7: Statistics of the generic graph datasets

| Dataset | type | Number of graphs | Number of nodes |
|---|---|---|---|
| Ego-small | Real | 200 | [4, 18] |
| Community-small | Synthetic | 100 | [12, 20] |
| Enzymes | Real | 587 | [10, 125] |
| Planar | Synthetic | 200 | 64 |
| Grid | Synthetic | 100 | [100,400] |

### I.4 DETAILS OF MOLECULE DATASETS

#### I.4.1 DATASET

**QM9**    It is a subset of the GDB-17 database and consists of 134,000 stable organic molecules, each containing up to 9 heavy atoms: carbon, oxygen, nitrogen, and fluorine (Ramakrishnan et al., 2014). The dataset includes 12 tasks related to quantum properties. We follow the train/test split from GDSS, using 12,000 molecules for training and the remaining 1,000 for testing.

**ZINC250k**    It contains 250,000 drug-like molecules with a maximum of 38 atoms per molecule (Irwin et al., 2012). It includes 9 atom types and 3 edge types. For a fair comparison, we use the same train/test split as previous works, such as GDSS and GSDM.

Table S8: Statistics of the molecular graph datasets

| Dataset | type | Number of graphs | Number of nodes | Number of node types | Number of edge types |
|---------|------|------------------|-----------------|----------------------|----------------------|
| QM9 | Real | 133,885 | [1, 9] | 4 | 3 |
| ZINC250k | Real | 249,455 | [6, 38] | 9 | 3 |

#### I.4.2 METRICS

For generic graph datasets, we employ Maximum Mean Discrepancy (MMD) to assess the distributions of graph statistics, specifically degree distribution, clustering coefficient, the number of occurrences of 4-node orbits, and eigenvalues of the normalized graph Laplacian. In alignment with prior research (Jo et al., 2022b), we utilize specialized kernels for MMD calculations: the Gaussian Earth Mover's Distance (EMD) kernel for degree distribution and clustering coefficient, the Gaussian Total Variation (TV) kernel for eigenvalues of the normalized graph Laplacian, and a standard Gaussian kernel for the 4-node orbits. To ensure a fair comparison, the size of the prediction set matches that of the test set.

**Validity**    We permit atoms to exhibit formal charges during valency checks because of the presence of formal charges in the training molecules. It is the fraction of valid molecules after valency correction or edge resampling.

**Validity w/o correction**    This metric explicitly evaluates the quality of molecule generation before any correction phase, providing a baseline for raw generation performance.

**FCD**    FCD quantifies the functional connectivity density within a molecule by computing distances and connectivity between atoms, based on both structural and chemical features. It describes the three-dimensional structure, topological features, and chemical properties of molecules, making it valuable in fields such as drug design, compound screening, and molecular simulations.

**NSPDK**    NSPDK assesses molecular similarity by comparing shortest paths within their graphical structures. It captures connectivity patterns and chemical environments, effectively describing relationships and similarities between molecules. For two distributions $p$ and $q$, the MMD using NSPDK is calculated as:

$$\text{MMD}^2_{\text{NSPDK}}(p, q) = \frac{1}{n(n-1)} \sum_{i=1}^{n} \sum_{j \neq i}^{n} k_{\text{NSPDK}}(\mathcal{X}_i, \mathcal{X}_j) + \frac{1}{m(m-1)} \sum_{i=1}^{m} \sum_{j \neq i}^{m} k_{\text{NSPDK}}(\mathcal{Y}_i, \mathcal{Y}_j) \quad (19)$$

$$- \frac{2}{mn} \sum_{i=1}^{n} \sum_{j=1}^{m} k_{\text{NSPDK}}(\mathcal{X}_i, \mathcal{Y}_j) \quad (20)$$

Here, $k_{\text{NSPDK}}(\cdot)$ denotes the NSPDK kernel function. $\mathcal{X}$ is the set of molecules from distribution $p$. $\mathcal{Y}$ is the set of molecules from distribution $q$. $n$ and $m$ represent the number of samples drawn from distributions $p$ and $q$, respectively. This formula quantifies the difference between the distributions $p$ and $q$ using the NSPDK kernel.

## I.5  DETAILS OF CONDITIONAL GENERATION

We included three guidance baselines in our conditional generation task:

**DiGress model with guidance**    Utilizing the guidance method integrated into the DiGress model (Vignac et al., 2022).

**Direct supervised training (ST)**    It involved selecting training samples from the dataset that satisfied $|\mu - \mu^*| < 1.0$ and retraining them using supervised learning settings identical to those in Section 4.2.

**Supervised fine-tuning (SFT)**    This method involved fine-tuning a pre-trained GGFlow model on molecules generated with $|\mu - \mu^*| < 1.0$, maintaining the same training settings as in Section 4.2.

These models were trained over 10,000 steps using the training settings detailed in Section 4.2. We then generated 1,000 samples to calculate the results for each guidance method and the unconditional method, with the values of $\mu$ estimated using Psi4 (Smith et al., 2020). We set the hyperparameters $\alpha$ and $\beta$ as 0.999 and 0.001.

# J  EXPERIMENT SETTINGS

## J.1  HYPERPARAMETER SETTINGS

Table S9 presents the hyperparameters employed in our experimental setup. For each dataset, the final resutls in Table 1 and Table 2 are the means of 5 different runs.

Table S9: Hyperparameter settings of different datasets

| Hyperparameter | Ego-small | Community-small | Grid | Planar | Enzymes | QM9 | ZINC250k |
|---|---|---|---|---|---|---|---|
| Number of layers | 5 | 7 | 5 | 4 | 6 | 9 | 9 |
| Hidden dimension of X | 256 | 256 | 256 | 256 | 256 | 256 | 128 |
| Hidden dimension of E | 128 | 128 | 128 | 128 | 128 | 128 | 64 |
| Hidden dimension of y | 128 | 128 | 128 | 128 | 128 | 128 | 64 |
| Optimizer | Adamw | Adamw | Adamw | Adamw | Adamw | Adamw | Adamw |
| Learning rate | $2 \times 10^{-4}$ | $2 \times 10^{-4}$ | $2 \times 10^{-4}$ | $2 \times 10^{-4}$ | $2 \times 10^{-4}$ | $2 \times 10^{-4}$ | $2 \times 10^{-4}$ |
| Batch size | 64 | 128 | 4 | 64 | 8 | 512 | 128 |
| Number of epochs | 2000 | 3000 | 5000 | 5000 | 10000 | 1000 | 1000 |
| Number of sampling steps | 500 | 500 | 500 | 500 | 500 | 500 | 500 |

## J.2  ABLATION STUDIES SETTINGS

For the evaluation of varying inference steps, we followed the same experimental settings as outlined in Sections 4.1 and 4.2. Samples were generated for 10 runs. The results were then visualized using the mean and variance across these 10 runs. It is important to note that in DiGress, the number of inference steps is constrained by its predefined diffusion steps ($N = 500$), so the DiGress curve terminates at 500 inference steps.

For the ablation studies of GGFlow without Optimal Transport (GGFlow w/o OT), GGFlow without GraphEvo (GGFlow w/o Evo) and GGFlow without GraphEvo and optimal transport (GGFlow w/o both), we adhered to the settings described in Sections 4.1 and 4.2. The final results were obtained by averaging the outcomes from five different runs.

To further investigate the advantages of optimal transport, we present generation results with varying inference steps on the Community-small and Planar datasets. As shown in Figure S2, GGFlow demonstrates superior generation quality compared to GGFlow (w/o OT), exhibiting narrower confidence intervals and comparable performance with fewer inference steps, which suggests that optimal transport enhances sampling both efficiency and stability.

We provide training loss and average values on Community-small datasets compared to DiGress, which shares the same training objectives. For fair comparisons, we use GGFlow (w/o both) and GGFlow (w/o Evo) to demonstrate the superiority of the flow matching framework and optimal

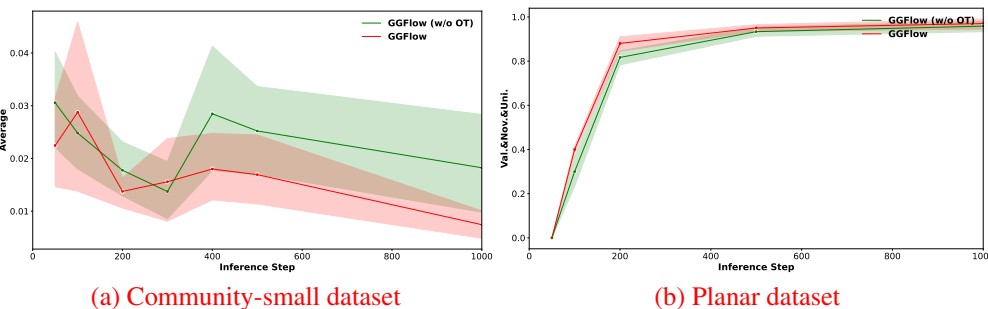

(a) Community-small dataset  (b) Planar dataset

Figure S2: Ablation studies of varying inference steps on Community-small and Planar datasets

transport. The average values are calculated as in Section 4.1 using checkpoints saved every 100 epochs.

Figure S3 demonstrates that GGFlow (w/o both) and GGFlow (w/o Evo) achieve faster and more effective convergence than DiGress. Furthermore, GGFlow (w/o Evo) outperforms both GGFlow (w/o both) and DiGress in average metrics, demonstrating the benefits of flow matching and optimal transport. These results indicate that optimal transport and flow matching framework contribute to greater training stability.

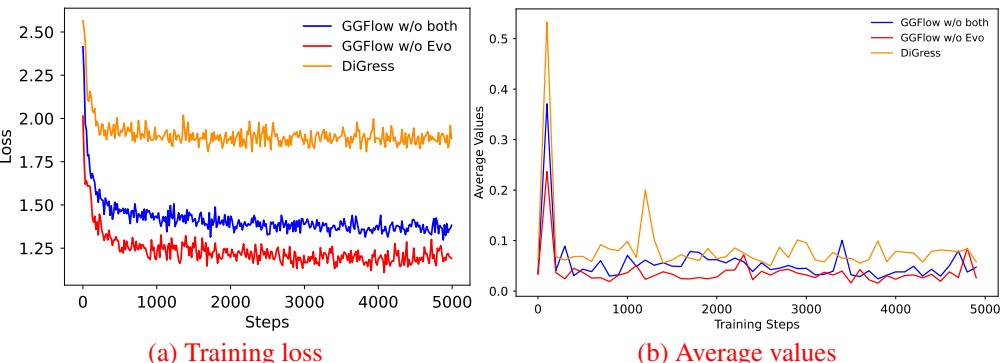

(a) Training loss  (b) Average values

Figure S3: Training stability analysis of flow matching and optimal transport in Community-small dataset

### J.3 GENERATION TIMES WITH DIFFERENT GRAPH SCALES

We conducted experiments to evaluate generation times across various graph scales using the Enzymes dataset. The GraphEvo model was configured with six layers, and generation time was measured for a single graph on an NVIDIA A100 80G GPU.

Table S10: Generation times with different graph scales.

| Number of Nodes | 10 | 50 | 100 | 200 | 400 |
|---|---|---|---|---|---|
| Time (s) | 1.92 | 4.24 | 12.15 | 48.59 | 235.8 |

## K VISUALIZATION

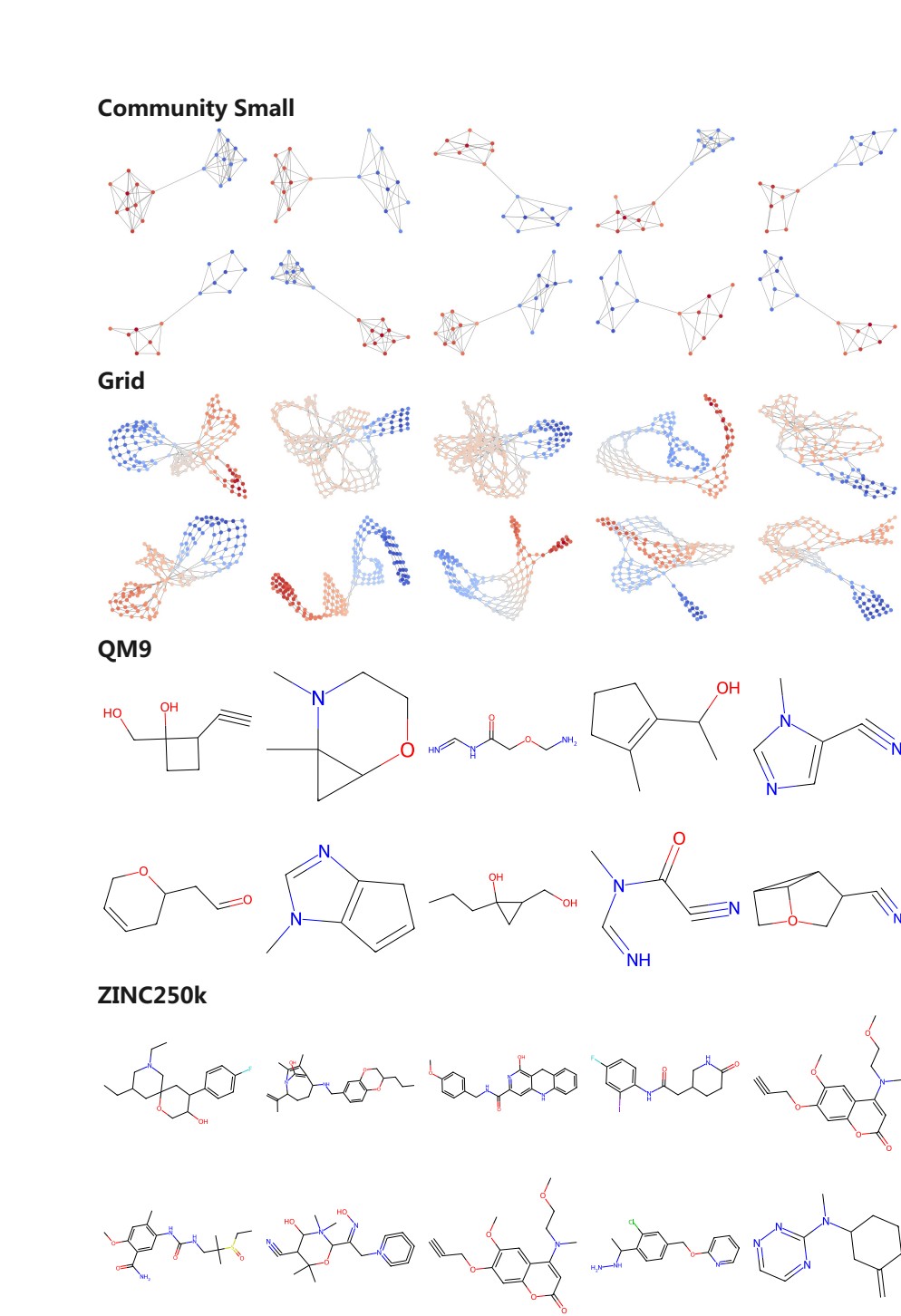

Figure S4: Visualization of generated samples of our model in different datasets

