# OpenReview forum: "Improving Graph Generation with Flow Matching and Optimal Transport"
_ICLR.cc/2025/Conference — Submitted to ICLR 2025_

### Official Review · Reviewer_PZdL · 2024-10-30

**Soundness:** 1
**Presentation:** 2
**Contribution:** 2
**Rating:** 1
**Confidence:** 5

**Summary:**

The paper introduces GGFlow, a graph generative model based on discrete flow matching and optimal transport. GGFlow aims to improve upon current graph generation methods, particularly diffusion-based models, by addressing two challenges: training stability and sampling efficiency. Additionally, GGFlow introduces a goal-guided generation framework using reinforcement learning, enabling conditional generation.

GGFlow generates graphs by transforming an initial noise distribution to a target data distribution through a smooth probability path, defined by a probability velocity field. This approach  the complexities associated with stochastic processes as in diffusion models.
To reduce the variance in training and sampling, GGFlow incorporates optimal transport to construct the joint distribution of the source and target graphs. The model integrates an edge-augmented transformer architecture (GraphEvo), which uses a triangle attention mechanism.

Using reinforcement learning, GGFlow can guide the graph generation process to meet predefined objectives, allowing for conditional generation where desired properties are targeted directly.
GGFlow demonstrates high performance on small graph generation tasks.

The article evaluates the model on synthetic and real-world datasets of small graphs, demonstrating effectiveness of the method. The experimental section includes an ablation study to assess the impact of both GraphEvo and Optimal Transport on model performance.

**Strengths:**

The paper is well-written and thoroughly documented. The appendix provides useful additional insights.

The proposed model introduces a novel graph generative approach based on discrete flow matching, which is a novel approach for graph generation.

The method demonstrates competitive generative performance on small graph datasets, showing its effectiveness and relevance.

These are concise but significant strengths.

**Weaknesses:**

Weaknesses after rebuttal:

### 1. OT is Not Permutation-Invariant

One of the claimed main contributions of the paper is the use of Optimal Transport (OT) for graphs. However, the OT employed is not permutation-invariant, making it an OT on adjacency matrices under specific permutations, rather than on graphs.

The OT relies on the Hamming distance, which aggregates element-wise differences between adjacency (and annotation) matrices. This element-wise computation is dependent on node ordering, meaning that permuting one matrix changes the distance: \( H(G_0, G_1) \neq H(G_0, \pi G_1) \).

While the lack of permutation invariance would not be a problem if directly acknowledged, the paper obscures this limitation, and some formulations are misleading.

---

### 2. Misleading Claims

Theorem 1 states: "the optimal transport map exhibits invariance under identical permutations." While this is technically true, it is misleading because it implies that the OT is invariant under all permutations, which is not the case. Specifically, if one matrix is permuted independently, the Hamming distance is not preserved: \( H(G_0, G_1) \neq H(G_0, \pi G_1) \).

This leads to other misleading claims:

- Lines 54–55: "The model preserves graph sparsity and permutation invariance, which is essential for realistic graph generation."
  - This is true without OT but incorrect with OT.

- Line 272: "Graphs are invariant to random node permutations, and GGFlow preserves this property."
  - Similarly, this is true without OT but incorrect with OT.

- Lines 275–277: "Since the source and target distributions are permutation invariant, the independent coupling also exhibits this invariance. Our optimal transport map, derived from Equation 7, similarly demonstrates invariance to identical permutations."
  - The use of "similarly" is misleading, as "invariance to identical permutations" is fundamentally different from true permutation invariance.

The paper does not address how this non-invariant OT relates to truly invariant graph distances, leaving a significant gap in the discussion.

---

### 3. Inconsistent Results and Interpretation

The authors claim that OT improves generative performance. However, the ablation study (Table 4) shows the model without OT performing better on 3 out of 8 metrics. With metrics being close and no standard deviations provided, there is no evidence of a significant effect of OT.

Additionally, in the enzyme dataset, the paper reports MMD distances between generated and test sets that outperform the MMD between training and test sets. The explanation provided is incorrect, as MMD, being a distance between distributions, does not directly depend on sample size.

---

### Summary

The paper's claims around OT, permutation invariance, and the effectiveness of OT lack clarity and accuracy. Furthermore, the experimental results do not convincingly support the proposed benefits of OT. These issues should be addressed to ensure transparency and scientific rigor.



___ Version before rebuttal ____

The paper raise some concerns:
- Some claims in the paper are misleading.
- Some results are questionable (generated graph distribution measured closer to the test set than the training set distribution?)


**Theorem 1 Validity**

The claim in Theorem 1 is problematic because the proof relies on the invariance of the Hamming distance under arbitrary permutations. However, the proof should show that the distance is invariant to independent permutations, i.e., $H(G^0, G^1) = H(\pi^0G^0, \pi^1G^1)$ instead of identical permutations (as exposed in appendix C.4: $H(G^0, G^1) = H(\pi G^0, \pi G^1)$). Therefore, the invariance does not hold. Consequently, if this assumption is indeed incorrect, then the Optimal Transport (OT) mechanism does not operate directly on graphs, but rather on specific graph representations, undermining a key contribution of the paper.

If Theorem 1 indeed fails, this calls into question the effectiveness of OT as applied in this context, and a significant part of the article should be reviewed.

**Experiments**

The experimental section has several limitations, and addressing these would significantly strengthen the work's impact. The main experimental limitations are as follows:

- Small Datasets: Except for the 'grid' dataset, which is a peculiar synthetic dataset, the model has only been tested on small graphs (up to 64 nodes for Planar, which is only reported in the appendix, and up to 38 nodes for Zinc). Testing on larger datasets such as SBM, Enzymes, Ego, or Proteins would improve the evaluation.

- Overfitting Concerns on Small Datasets: Evaluations on larger datasets are especially important given that generic datasets, often containing few instances, can easily be overfit. For instance, the lower novelty and uniqueness score on Ego-Small suggests some degree of overfitting on that dataset.

- Incomplete Ablation Study: While the model introduces several additional components, a more systematic ablation study on multiple dataset would clarify each element’s contribution. Moreover, a comparison between discrete flow matching and discrete diffusion would highlight the advantages of the flow matching framework. To ensure fairness, it would be beneficial to use a discrete diffusion model with identical architecture, extra features, and hyperparameters as a baseline. For molecular generation, presenting results without additional molecular features (which are not used in most baseline models and has been improved significantly the molecular metrics) would also be insightful.

- Scalability: While scalability is briefly mentioned as model limitation in the conclusion, it would be beneficial to include indicators of this issue in the experimental section. For instance, reporting generation times would provide insight about the model scalability.

**Lack of Novelty/Originality**

The paper represents an application of discrete flow matching to graph generation. This type of approach was expected; indeed, another paper submitted paper proposes a similar contribution (https://openreview.net/forum?id=ZGRRC514rI).

**Missing Explicit References in Method Presentation**

Section 3 would benefit from more explicit references to foundational works. For instance, it would be useful to specify which discrete flow matching frameworks, as cited in Section 2, the authors drew inspiration from and how GGFlow differs from these.

**Minor Comments**

Line 51: This sentence lacks a main verb.

Line 44: Could you clarify the instability issues encountered in training diffusion models or provide relevant references?

**Questions:**

My evaluation relies primarily on the accuracy of Theorem 1 and the experimental design. The current rating assumes that Theorem 1 is incorrect. Naturally, if further clarification or evidence proves this theorem valid the rating would be significantly revised.
Similarly, clarification and/or improvement to the experimental section would change my evaluation.

---

> ### Author Response · Authors · 2024-11-19
>
> We sincerely appreciate the reviewer’s constructive feedback. We have carefully addressed each comment and made the necessary revisions accordingly.
>
> **Q: Validity of Theorem 1**
>
> A: In our method, we assume that all pairs between the source and target data share the same node order, as the interpolation between them relies on pairs with identical node order. During the training procedure, our network $\hat{p}^1(G^1|G^0,G^t)$ maintains permutation equivariance, such that $\hat{p}^1(G^1|\pi G^0,\pi G^t) = \pi \hat{p}^1(G^1| G^0, G^t)$ for any permutation $\pi$ to approximate $\pi G^1$. Additionally, the interpolation $\pi G^t = t\pi G^0 +(1-t)\pi G^1$ necessitates that the $G^0$ and $G^1$ share identical permutations, thus we prove the invariance of optimal transport under identical permutations, i.e. $\phi(G^0,G^1)=\phi(\pi G^0,\pi G^1)$.
>
> Since the prior distribution is permutation invariant and our GraphEvo model is permutation equivariant, the generated samples also maintain permutation invariance, as established in Lemma 1, which adapts Theorem 5 from CatFlow [1] to our framework in Appendix C.4 of our latest submission. We have added these clarifications in lines 986-992 of Appendix C.4.
>
> Moreover, our model adopts the Hamming distance to balance the efficiency and performance of optimal transport. The results in our paper can demonstrate the utility of our optimal transport and we additionally tested the computation time of optimal transport on different datasets to demonstrate its efficiency. These results are included in lines 1084-1101 of Appendix E in our latest submission.
>
> |  | Training Step | OT Time | Percentage | Model Size | Batch Size | Number of nodes |
> | --- | --- | --- | --- | --- | --- | --- |
> | Planar | 0.126349 | 0.0024997 | 1.9% | 3.6 M    | 64 | 64 |
> | Zinc250K | 0.130121 | 0.00699354 | 5.3% | 4.6 M | 128 | [6,38] |
> | Community Small | 0.04079 | 0.00236716 | 5.6% | 6.4 M | 80 | [12,20] |
>
> **Q: Results in Enzymes dataset**
>
> A: We have included the results for the Enzymes dataset in Table S4 of Appendix H in our latest submission. These results further demonstrate GGFlow's superiority in larger graph datasets.
>
> | Method | Deg. | Clus. | Orbit | Avg. |
> | --- | --- | --- | --- | --- |
> | EDP-GNN | 0.023 | 0.268 | 0.082 | 0.124 |
> | GDSS | 0.026 | 0.102 | 0.009 | 0.046 |
> | GSDM | 0.013 | 0.088 | 0.010 | 0.037 |
> | DiGress | 0.010 | 0.046 | 0.002 | 0.019 |
> | GGFlow | 0.008 | 0.026 | 0.002 | 0.012 |
>
> **Q: Overfitting Concerns on Ego Small Dataset**
>
> A: We acknowledge that many methods, including ours, may face overfitting risks on the Ego Small dataset due to its simple graph structure and limited data. However, the superior performance of our method on larger graph datasets, such as Enzymes, Planar, and Grid, demonstrates its effectiveness.
>
> **Q:  Ablation Study**
>
> A: In Table 4 of the initial manuscript, GGFlow (w/o GraphEvo) and GGFlow (w/o both), which shares the same architecture, extra features, and hyperparameters as DiGress, demonstrate superior performance, highlighting the advantages of the flow matching framework over the diffusion framework.
>
> Following your suggestion, we conducted an additional ablation study on the Planar dataset. The results indicate that all components contribute to our strong performance, which are included in Table 4 of Section 4.4 in our latest submission.
>
> | Method | Deg. | Clus. | Orbit | Spec. | Val.&Nov.&Uni. |
> | --- | --- | --- | --- | --- | --- |
> | DiGress | 0.0003 | 0.0372 | 0.0098 | 0.0106 | 87.5 |
> | GGFlow (w/o both) | 0.0023 | 0.1076 | 0.0053 | 0.0099 | 92.5 |
> | GGFlow (w/o OT) | 0.0015 | 0.0431 | 0.0020 | 0.0067 | 97.0 |
> | GGFlow (w/o Evo) | 0.0020 | 0.0763 | 0.0034 | 0.0124 | 94.5 |
> | GGFlow | 0.0156 | 0.0196 | 0.0019 | 0.0091 | 97.5 |
>
> Additionally, we focus on results without extra features, labeled GGFlow (w/o extra), on the Zinc250k dataset due to the limited time and computational resources, demonstrating that these features enhance performance. DiGress also demonstrated the utility of these features in Table 6 of their paper [2].
>
> | ZINC250K | Valid | Valid. w/o corr. | NSPDK | FCD |
> | --- | --- | --- | --- | --- |
> | DiGress | 100 | 94.98 | 0.0021 | 3.482 |
> | GGFlow (w/o extra) | 100 | 98.03 | 0.0012 | 1.667 |
> | GGFlow | 100 | 99.63 | 0.0010 | 1.455 |
>
>
>
> [1] Eijkelboom, Floor, et al. "Variational Flow Matching for Graph Generation." arXiv preprint arXiv:2406.04843 (2024).
>
> [2] Vignac, Clement, et al. "DiGress: Discrete Denoising diffusion for graph generation." *The Eleventh International Conference on Learning Representations*.

---

> > ### Comment · Reviewer_PZdL · 2024-11-24
> >
> > Thanks you for your answer. I will focus here on the correctness of theorem 1, because it is the most problematic.
> >
> > I am still think that your theorem 1 is incorrect. I will explain my view numerating the affirmation so you can easily point out where you disagree.
> > 1. I agree that once the graphs are coupled, you method is equivariant. My point concerns only the computation of the optimal transport map.
> > 2. The optimal transport map is invariant only if the Hamming distance is invariant.
> > 3. A distance $D$ between graph is invariant only if $D(G_1, G_2) = d \implies D(G_1, \pi G_2) = d$
> > 4. The Hamming distance is not invariant as defined in 3.
> > 5. Because of 4 your optimal transport map is not permutation-invariant.
> > I would appreciate that you show me were I am wrong. It is also the better way the convince me.
> >
> > Now some additional remarks and questions:
> > 6. I agree that $\phi(G_1, G_2) = \phi(\pi G_1, \pi G_2)$ but this is not the way of ensuring invariance of a distance
> > 7. 'Thus, we prove the invariance of optimal transport under identical permutations'. Yes, but you should prove it under any permutation.
> > 8. I am not sure what you mean by 'we assume an node ordering'.
> > 9. What is this node ordering?
> > 10. On what rely this assumption?
> >
> > I am looking forward to reading from you

---

> ### Author Response · Authors · 2024-11-19
>
> **Q: Scalability**
>
> A: We conducted experiments to evaluate generation times across various graph scales using the Enzymes dataset. The GraphEvo model was configured with six layers, and generation time was measured for a single instance on an NVIDIA A100 80G GPU. The results are detailed in lines 1455-1457 of Appendix J.3 in our latest submission.
>
> | Number of nodes | 10 | 50 | 100 | 200  | 400 |
> | --- | --- | --- | --- | --- | --- |
> | Time | 1.92s | 4.24s | 12.15s | 48.59s | 235.8s |
>
> **Q: Lack of novelty**
>
> A: The DeFoG model was released after our submission, so we believe it does not diminish the novelty of our work. Furthermore, our main contributions include the integration of optimal transport into flow matching, the development of a new architecture, and the implementation of goal-directed conditional generation. We are confident that our method will make a contribution to the community.
>
> **Q: Missing Explicit References in Method Presentation**
>
> A: Thank you for your suggestion. We have added the relevant section in lines 861-870 of Appendix B.2 in our latest submission.
>
> **Q: Minior1**
>
> A: We have corrected this issue.
>
> **Q: Minior 2**
>
> A: Due to limited time and computing resources, we were unable to reproduce the training instability of diffusion models. However, several references discuss this phenomenon [1].
>
> [1] Karras, Tero, et al. "Analyzing and improving the training dynamics of diffusion models." *Proceedings of the IEEE/CVF Conference on Computer Vision and Pattern Recognition*. 2024.

---

> ### Author Response · Authors · 2024-11-22
>
> Dear Reviewer,
>
> Thank you very much for your thoughtful review. We have addressed the concerns you raised and look forward to your constructive feedback.

---

> ### Author Response · Authors · 2024-11-25
>
> Thank you for your reply, and we apologize for any unclear presentation.
>
> We acknowledge that your statements (2-5) are correct. However, in our method settings, we only need to ensure optimal transport invariance under identical permutations, i.e. $\phi (G^0,G^1) = \phi (\pi G^0,\pi G^1)$ for any permutation $\pi$. We would like to clarify the following points:
>
> 1. **Purpose of Theorem 1:** Our Theorem 1 aims to prove the permutation invariance of the optimal transport map for the same permutation. Given our objectives, the theorem holds true.
>
> 2. **Reasons for Same Permutation:** We will explain why our method demonstrates permutation invariance specifically for the same permutation.
>
> ## Reasons:
>
> The goal of optimal transport is to pair source and target data points with minimal cost during training, which is beneficial for our interpolation [1,2]. Thus, we design our optimal transport approach from the perspective of interpolation. The interpolation of source and target data is defined as $G^t = tG^0 + (1-t)G^1$.
>
> **(What is the node order?)** It is important to clarify the node order of the graph $G$: If the node set of $G$ is $\\{A,B,C\\}$, the possible node orders include $(A,B,C)$ or $(B,A,C)$ or $(C,B,A)$ and is the order for our nodes and edges in matrix representation.
>
> 1. In the interpolation process, we transform the graph representation to a matrix representation before performing interpolation. For example, for source data $G^0=(V^0, E^0), V^0\in \mathbb{R}^{a\times n}, E^0\in \mathbb{R}^{a\times a\times  m}$ and target data $G^1=(V^1, E^1), V^1\in \mathbb{R}^{a\times n}, E^1\in \mathbb{R}^{a\times a\times  m}$, where $a$ is the number of nodes, $n$ is the class number of nodes, and $m$ is the class number of edges, the node orders of $G^0$ and $G^1$ have been fixed. Therefore, interpolation is performed directly on these fixed node orders.
>
> 2. **(On what relies this assumption?)** The optimal transport aims to find pairs with the minimum cost for interpolation, and interpolation is conducted on a fixed node order. Additionally, during optimal transport calculations, we also utilize the matrix representation of these graphs and our prior distribution is permutation invariant. Therefore, our objective is to match source data with the target data $G^1$ whose node order is fixed, to achieve minimal transport cost. Furthermore, we assume that all pairs of source and target data share the same node order during optimal transport, which also facilitates the identification of pairs with minimal cost.
>
>
> 3. **(Why we need to prove Theorem 1?)** Regarding the permutation of the intermediate graph $G^t$, we have $\pi G^t = t\pi G^0 + (1-t)\pi G^1$, where $G^0$ and $G^1$ share an identical permutation. Our network $\hat{p}^1(G^1|G^0,G^t)$ needs to maintain permutation equivariance, such that $\hat{p}^1(G^1|\pi G^0,\pi G^t) = \pi \hat{p}^1(G^1| G^0, G^t)$ for any permutation $\pi$ to approximate $\pi G^1$. So we prove the invariance of optimal transport under identical permutations, i.e. $\phi(G^0,G^1)=\phi(\pi G^0,\pi G^1)$.
>
> 4. **(What is Theorem 1)** Considering the invariance of Hamming distance to the same permutation, i.e. $H(G^0, G^1) = H(\pi G^0, \pi G^1)$ for any permutation $\pi$, our optimal transport is permutation invariance to same permutation i.e. $\phi(G^0,G^1)=\phi(\pi G^0,\pi G^1)$.
>
>
> **(Why Choose This Method Setting?)** Moreover, we have tested the cost with invariance to any permutation in our initial experiments. However, computational methods such as the GOT Distance [3] and FGW distance [4] have prohibitive computational costs. Therefore, we opted to utilize the Hamming distance as our cost metric, enabling us to achieve our objectives with acceptable computational efficiency.
>
> In summary, Theorem 1 is designed to ensure the invariance of optimal transport under identical permutations. According to our experimental results, this invariance benefits our training process and generation results.
>
> We hope this explanation clarifies Theorem 1 and addresses your concerns.
>
>
> [1] Bose, Avishek Joey, et al. "SE(3)-stochastic flow matching for protein backbone generation." arXiv preprint arXiv:2310.02391 (2023).
>
> [2] Song, Yuxuan, et al. "Equivariant flow matching with hybrid probability transport for 3d molecule generation." Advances in Neural Information Processing Systems 36 (2024).
>
> [3] Chen, Liqun, et al. "Graph optimal transport for cross-domain alignment." International Conference on Machine Learning. PMLR, 2020.
>
> [4] Titouan, Vayer, et al. "Optimal transport for structured data with application on graphs." International Conference on Machine Learning. PMLR, 2019.

---

> ### Comment · Reviewer_PZdL · 2024-11-25
>
> 'Theorem 1:
> If the distributions $p(G_0)$ and $p(G_1)$ are permutation invariant and the cost func-
> tion maintains this invariance, then the optimal transport map ϕ also respects this property, i.e.,
> $ϕ(G_0, G_1) = ϕ(πG_0, πG_1)$, where π is a permutation operator.'
>
> In the graph learning community, permutation-invariance refers to invariance under any permutation of the input node ordering.
>
> Regarding Theorem 1, the following statements seem implied:
>
> 1. The cost function is permutation-invariant with respect to any permutation of the input node ordering.
> 2. Consequently, the optimal transport map is also permutation-invariant with respect to any permutation of the input node ordering.
> 3. An optimal transport map \( \phi \) is permutation-invariant with respect to any permutation of the input node ordering, if \( \phi(G_0, G_1) = \phi(\pi G_0, \pi G_1) \), where \( \pi \) is a permutation.
>
> However, as acknowledged, these three statements are incorrect. The current formulation is, at the very least, misleading.
>
> Additionally, the confusion is compounded by the preceding statement:
> "To optimize the computational efficiency of OT and preserve permutation invariance, we define the distance..."
>
> Theorem 1 should be reformulated to eliminate confusion, as it is incorrect to state that the cost function and the transport map are permutation-invariant (implicitly understood with respect to any permutation of the input node ordering).

---

> ### Author Response · Authors · 2024-11-26
>
> Thank you for your patient and constructive feedback. We have made the necessary revisions to Theorem 1 and its clarifications in accordance with your suggestions to avoid any misleading information. **Theorem 1 is now presented as follows:**
>
> _If the distributions $p(G^0)$ and $p(G^1)$ are permutation invariant, and the cost function maintains invariance under identical permutations, i.e., $H(G^0, G^1) = H(\pi G^0, \pi G^1)$ for any permutation $\pi$, then the optimal transport map $\phi$ also exhibits invariance under identical permutations, such that $\phi(G^0, G^1) = \phi(\pi G^0, \pi G^1)$._
>
> Additionally, we have revised the sentences in lines 212 and 275-282 of our latest submission to avoid any potential confusion for readers. We have also added clarifications from our discussion regarding our method settings in lines 989-1024 of Appendix C.4 to provide a more detailed explanation of the rationale behind Theorem 1. We also hope these clarifications will also inspire other researchers in the field. All of these revisions are marked in red in our latest submission.
>
> We appreciate your valuable feedback and suggestions, which have significantly improved the quality of our manuscript. We sincerely hope that our new submission and revisions have addressed your concerns.

---

> > ### Comment · Reviewer_PZdL · 2024-11-28
> >
> > Despite your corrections, I still believe your article relies on misleading claims, such as: "we propose (...) a generative model incorporating optimal transport for graph structure", which seems implying that your method implement a proper OT between graphs.
> >
> > More generally, most of my concerns remain unaddressed.
> >
> > ### Regarding Optimal Transport (OT):
> > - What does it mean to perform OT on specific graph representations (under a specific ordering)?
> > - How would the coupling differ if a permutation-invariant distance were used instead of the Hamming distance?
> > - You claim that GGFlow outperforms GGFlow w/o OT, but I interpret the results differently. According to Table 4, I would argue they are on par.
> > - Given the unclear effect of OT, the title of your paper seems misleading.
> > - OT, based on a real, permutation-invariant graph distance, would have been a significant contribution. Without it, the paper's contribution feels limited.
> >
> > ### Regarding the Experiments:
> > - My concerns about the molecular extra features you use remain unaddressed (the table for GGFlow w/o extra features does not appear in the paper).
> > - Your additional experiment on Enzymes raises questions. How do you explain that your model outperforms... the training set?
> > - As a consequence of the above points, along with my concerns regarding Ego and Grid (see review), the evaluation of your model relies heavily on Planar and Community Small, where its superior performance is not obvious.

---

> > > ### Author Response · Authors · 2024-11-29
> > >
> > > ## Regarding Optimal Transport
> > >
> > > 1. We have added detailed explanations and rationales about our optimal transport in the prior discussion and Appendix C.4. According to Table 4, Figure S2, and Figure S3, we believe that our optimal transport benefits the training and inference efficiency of our model. It also ensures the permutation property of generated graphs.
> > >
> > > 2. Most permutation invariant distances [1,2] have $O(n^2m^2)$ or $O(n^2m+mn^2)$ time complexity, where $n$ is node number and $m$ is edge number. The time consumption of optimal transport is intolerable using these cost functions during the training procedure.
> > >
> > > 3. In our experimental settings, GGFlow (w/o Evo) and GGFlow (w/o both) share the same architecture, extra features, and hyperparameters as DiGress [3]. The results in Table 4 of our submission demonstrate that GGFlow (w/o both) outperforms DiGress across most metrics, highlighting the advantages of the flow matching framework over the diffusion framework in graph space. Moreover, GGFlow (w/o Evo) also outperforms GGFlow (w/o both) in nearly all metrics, indicating the effectiveness of optimal transport. Notably, in the Planar dataset, the Val.&Nov.&Uni. metric of GGFlow (w/o Evo) has improved by 8% compared to the previous SOTA model.
> > >
> > >     Additionally, we provided training analysis in Appendix J.2. According to Figure S3, GGFlow (w/o both) and GGFlow (w/o Evo) show better training convergence and stability than DiGress. During the training procedure, GGFlow (w/o Evo) demonstrates better and more stable generation results than GGFlow (w/o both), indicating that optimal transport significantly improves the training stability and efficiency of our model.
> > >
> > > 4. We believe that our optimal transport is a primary contribution, as it enhances our model's training and inference performance and is efficient and effective compared to other graph distances. Moreover, our optimal transport preserves the permutation property of generated graphs, as shown in Lemma 1 in Appendix C.4. We also revise our Theorem 1 to avoid any misleading.
> > >
> > > ## Regarding Experiments
> > >
> > > 1. We provide ablation studies of extra features in our model on the Zinc250k dataset to demonstrate their utility in our first rebuttal. However, these extra features are not a primary contribution of our models; we have clarified in our initial submission that they come from DiGress [3] and also cited DiGress in line 247 of our initial submission. Therefore, we did not provide ablation studies of these extra features in our submission.
> > >
> > > 2. In the Enzymes experiment, the results of the training set are calculated by the distance metric between the training set and the test set, while the results of our model and other methods are calculated by the distance between the generated results and the test set. The training and test sets are divided into a 4:1 ratio. The number of training and test sets is quite different; at the same time, the generated data used in the test is the same as the number of test sets. Therefore, the distance between the generated data from our model, DiGress [3], GSDM [4], and GDSS [5] and the test set is smaller than the distance between the training set and the test set, which does not mean that our results are problematic.
> > >
> > > 3. Our model achieves superior performance on various datasets, including generic and molecular datasets. We provide complete results in Tables 1, 2, 3, S2, S3, S4, S5, and S6, which demonstrate the superiority of our model. We believe that these results are sufficient to support the superior performance of our method and the contributions of our work.
> > >
> > >
> > > [1] Abu-Aisheh, Zeina, et al. "An exact graph edit distance algorithm for solving pattern recognition problems." 4th International Conference on Pattern Recognition Applications and Methods 2015. 2015.
> > >
> > > [2] Titouan, Vayer, et al. "Optimal transport for structured data with application on graphs." International Conference on Machine Learning. PMLR, 2019.
> > >
> > > [3] Vignac, Clement, et al. "DiGress: Discrete Denoising diffusion for graph generation." *The Eleventh International Conference on Learning Representations*.
> > >
> > > [4] Luo, Tianze, Zhanfeng Mo, and Sinno Jialin Pan. "Fast graph generation via spectral diffusion." IEEE Transactions on Pattern Analysis and Machine Intelligence (2023).
> > >
> > > [5] Jo, Jaehyeong, Seul Lee, and Sung Ju Hwang. "Score-based generative modeling of graphs via the system of stochastic differential equations." International conference on machine learning. PMLR, 2022.

---

### Official Review · Reviewer_aYqJ · 2024-11-02

**Soundness:** 2
**Presentation:** 3
**Contribution:** 2
**Rating:** 6
**Confidence:** 2

**Summary:**

This work proposes a discrete flow matching generative model incorporating optimal transport for graph structures with improved training stablility and efficient sampling compared with diffusion model. It achieves outstanding performance on both unconditional and conditional molecule graph generation tasks.

**Strengths:**

1. Clear illustration of graph flow matching method.
2. Strong performance on generation tasks. Clear ablation study.

**Weaknesses:**

1. Incomplete ablation study. Ablation of OF and GraphEvo is only conducted on one synthetic datasets.
2. Training stablility, as one of the benefits this work claims, is not compared in experiments.

**Questions:**

See weakness.

---

> ### Author Response · Authors · 2024-11-19
>
> We sincerely appreciate the reviewer’s constructive feedback. We have carefully addressed each comment and made the necessary revisions accordingly.
>
> **Q: Ablation studies of Planar dataset**
>
> A: Following your suggestion, we conducted an additional ablation study on the Planar dataset. The results indicate that all components contribute to our strong performance, which have been included in Table 4 of Section 4.4 in our latest submission.
>
> | Method | Deg. | Clus. | Orbit | Spec. | Val.&Nov.&Uni. |
> | --- | --- | --- | --- | --- | --- |
> | DiGress | 0.0003 | 0.0372 | 0.0098 | 0.0106 | 87.5 |
> | GGFlow (w/o both) | 0.0023 | 0.1076 | 0.0053 | 0.0099 | 92.5 |
> | GGFlow (w/o OT) | 0.0015 | 0.0431 | 0.0020 | 0.0067 | 97.0 |
> | GGFlow (w/o Evo) | 0.0020 | 0.0763 | 0.0034 | 0.0124 | 94.5 |
> | GGFlow | 0.0156 | 0.0196 | 0.0019 | 0.0091 | 97.5 |
>
> **Q: Training stability**
>
> We provide training loss and average values on community small datasets, comparing our method with DiGress, which shares the same training objectives. For fair comparisons, we analyze GGFlow (w/o both) and GGFlow (w/o Evo) to highlight the advantages of the flow matching framework and optimal transport. These results are detailed in our submission (lines 1447-1457) and illustrated in Figure S3 of Appendix J.2.
>
> Our results in Figure S3 indicate that GGFlow (w/o both) and GGFlow (w/o Evo) achieve faster and more effective convergence than DiGress using the same architecture. Furthermore, GGFlow (w/o Evo) surpasses both GGFlow (w/o both) and DiGress in average metrics, demonstrating the benefits of flow matching and optimal transport. These results suggest enhanced stability during training.

---

> ### Author Response · Authors · 2024-11-22
>
> Dear Reviewer,
>
> Thank you very much for your thoughtful review. We have addressed the concerns you raised and look forward to your constructive feedback.

---

### Official Review · Reviewer_nffN · 2024-11-03

**Soundness:** 2
**Presentation:** 3
**Contribution:** 2
**Rating:** 6
**Confidence:** 4

**Summary:**

This work introduces a discrete flow matching model for graph generation that uses optimal transport and edge-augmented graph transformer. The authors also introduce RL-based conditional graph generation. The proposed approach yields stable training and improves sampling efficiency.

**Strengths:**

- The discrete flow matching for graph generation seems to be novel, although there is existing work on flow matching framework for graph generation.

- The edge-augmented graph transformer seems to be an improvement of the previous graph transformer architecture.

- The motivation for using optimal transport to improve sampling efficiency for graph generation is reasonable but lacks experimental results.

**Weaknesses:**

- CatFlow (Eijkelboom et al., 2024) is a graph generation method based on variational flow matching, which is highly related to this work. Although there is a comparison in Appendix B.3, there should be an experimental comparison and explanation of why GGFlow is better.

- Experiments for generic graph generation should use larger datasets like Planar or SBM instead of Ego-small and Community-small datasets which consist of very small graphs. Additionally, validity metrics such as V.U.N. (valid, unique, and novel) should also be used to evaluate models as MMDs are not a reliable metric, especially for small graphs. In particular, for the Grid dataset, the validity metric should be used to show that the generative model can actually produce a grid structure.

- Ablation study should be also conducted using datasets where validity or similar metrics can be measured, e.g., Planar or SBM datasets.  Relying only on MMD does not give much information on how the performance is improved.

- Comparison on molecule graph generation is not fair. It is not clear whether the new graph transformer architecture or the flow matching framework provides the performance improvement. In order to show this, there should be an ablation study on using the same architecture.

- The ablation study on sampling efficiency improvement based on optimal transport is not clear. Which experiments support this claim? Figure 3 only shows that GGFlow outperforms DiGress or GDSS, not the OT ablation.


[1] Eijkelboom et al., Variational Flow Matching for Graph Generation, 2024

**Questions:**

Please address the weakness above.

---

> ### Author Response · Authors · 2024-11-19
>
> We sincerely appreciate the reviewer’s constructive feedback. We have carefully addressed each comment and made the necessary revisions accordingly.
>
> **Q: Comparison with CatFlow**
>
> A: Since CatFlow [1] has not made its source code available, we did not include it in our experiments for fairness. Instead, we clarified the differences between our methods in Appendix B.3.
>
> The tables below present results directly from their manuscript, where our methods demonstrate superior performance.
>
> | Ego Small | Degree | Clustering | Orbit |
> | --- | --- | --- | --- |
> | CatFlow | 0.013 | 0.024 | 0.008 |
> | GGFlow | 0.005 | 0.033 | 0.004 |
>
> | Community Small | Degree | Clustering | Orbit |
> | --- | --- | --- | --- |
> | CatFlow | 0.018 | 0.086 | 0.007 |
> | GGFlow | 0.011 | 0.030 | 0.002 |
>
> Three reasons contribute to our better performance:
>
> 1. **Model Architecture**: CatFlow employs a graph transformer similar to DiGress, whereas our method utilizes an edge-augmented graph transformer to improve edge communication.
> 2. **Prior Distribution**: CatFlow uses a standard normal distribution as its prior, while our reference distribution is specifically designed to closely approximate the true data distribution, addressing graph sparsity.
> 3. **Source and Target Coupling**: CatFlow implements independent source and target coupling, while our method employs optimal transport to extend coupling to more general scenarios. This approach results in straighter generative trajectories, thereby enhancing performance.
>
> **Q: Results of Planar dataset and validity of Grid dataset**
>
> A: We have included the Planar dataset in our initial manuscript, presenting the V.U.N. results in Table S2 of Appendix G. These results demonstrate the superiority of our models compared to other methods, including the V.U.N. metric.
>
> Previous works [2, 3, 4, 5] did not test validity for the Grid dataset, so we didn't report the metric in our initial submission. Following your suggestion, we calculated the validity metric for the Grid dataset using the method from the DiGress GitHub repository ([https://github.com/cvignac/DiGress/blob/9f526a05c8c55715d5c92b6a35d32ed758a9098b/src/analysis/spectre_utils.py#L581](https://github.com/cvignac/DiGress/blob/9f526a05c8c55715d5c92b6a35d32ed758a9098b/src/analysis/spectre_utils.py#L581)). The results, shown in the following table, indicate that our method achieves comparable validity than other approaches. However, the rationale for the validity calculation method remains unproven, as it relies on searching for isomorphic graphs across all grid data, including the training dataset, rather than checking features (such as planarity) of the generated graph to assess validity, as is done with the Planar dataset.
>
> | Method | SwinGNN | GDSS | DiGress | GGFlow |
> | --- | --- | --- | --- | --- |
> | Valid | 0.30 | 0.15 | 0.00 | 0.15 |
>
> **Q: Ablation studies in Planar dataset**
>
> A: Following your suggestion, we conducted an additional ablation study on the Planar dataset and reported the V.U.N. metric. The results indicate that all components contribute to our strong performance, which have been included in Table 4 of Section 4.4 in our latest submission.
>
> | Method | Deg. | Clus. | Orbit | Spec. | Val.&Nov.&Uni. |
> | --- | --- | --- | --- | --- | --- |
> | DiGress | 0.0003 | 0.0372 | 0.0098 | 0.0106 | 87.5 |
> | GGFlow (w/o both) | 0.0023 | 0.1076 | 0.0053 | 0.0099 | 92.5 |
> | GGFlow (w/o OT) | 0.0015 | 0.0431 | 0.0020 | 0.0067 | 97.0 |
> | GGFlow (w/o Evo) | 0.0020 | 0.0763 | 0.0034 | 0.0124 | 94.5 |
> | GGFlow | 0.0156 | 0.0196 | 0.0019 | 0.0091 | 97.5 |
>
> **Q: Results of molecule datasets using the same architecture**
>
> A: Given the limited time and computational resources, we focused on results from the Zinc250k dataset, employing the same model architecture as DiGress. In the table, GGFlow (w/o both) outperforms DiGress across nearly all metrics, demonstrating the superiority of our flow matching framework.
>
> | ZINC250K | Valid | Valid. w/o corr. | NSPDK | FCD |
> | --- | --- | --- | --- | --- |
> | DiGress | 100 | 94.98 | 0.0021 | 3.482 |
> | GGFlow (w/o both) | 100 | 97.34 | 0.0019 | 2.101 |
> | GGFlow | 100 | 99.63 | 0.0010 | 1.455 |
>
>
> [1] Eijkelboom, Floor, et al. "Variational Flow Matching for Graph Generation." arXiv preprint arXiv:2406.04843 (2024).
>
> [2] Jo, Jaehyeong, Seul Lee, and Sung Ju Hwang. "Score-based generative modeling of graphs via the system of stochastic differential equations." International conference on machine learning. PMLR, 2022.
>
> [3] You, Jiaxuan, et al. "Graphrnn: Generating realistic graphs with deep auto-regressive models." International conference on machine learning. PMLR, 2018.
>
> [4] Yan, Qi, et al. "Swingnn: Rethinking permutation invariance in diffusion models for graph generation." *arXiv preprint arXiv:2307.01646* (2023).
>
> [5] Luo, Tianze, Zhanfeng Mo, and Sinno Jialin Pan. "Fast graph generation via spectral diffusion." IEEE Transactions on Pattern Analysis and Machine Intelligence (2023).

---

> ### Author Response · Authors · 2024-11-19
>
> **Q: Ablation studies of optimal transport**
>
> A: The y-axis in Figure 3 has been scaled by DiGress and GDSS, making it difficult to observe the difference between GGFlow (w/o OT) and GGFlow. To address this, we have included an additional figure in Figure S2 (a) of Appendix J.2, which focuses solely on GGFlow and GGFlow (w/o OT). This figure demonstrates that GGFlow consistently achieves better average performance and narrower confidence intervals than GGFlow (w/o OT) across nearly all inference steps.
>
> Additionally, we conducted experiments on the Planar dataset to further illustrate the sampling efficiency of optimal transport in Figure S2 (b) of Appendix J.2. The results show that GGFlow outperforms GGFlow (w/o OT) in V.U.N. performance at all inference steps and can achieve comparable performance with fewer steps, reinforcing the superiority of our optimal transport approach.

---

> ### Author Response · Authors · 2024-11-22
>
> Dear Reviewer,
>
> Thank you very much for your thoughtful review. We have addressed the concerns you raised and look forward to your constructive feedback.

---

> > ### Comment · Reviewer_nffN · 2024-11-25
> > **Thank you for the response**
> >
> > Thank you for the additional experiments and explanations.
> > The authors have addressed the concerns, so I raise my score to 6.

---

> > > ### Author Response · Authors · 2024-11-25
> > >
> > > We sincerely appreciate the reviewer's insightful and constructive comments, which have significantly enhanced the quality of our manuscript.

---

### Official Review · Reviewer_Exo3 · 2024-11-04

**Soundness:** 3
**Presentation:** 3
**Contribution:** 2
**Rating:** 8
**Confidence:** 5

**Summary:**

The paper introduces GGFlow, a novel generative model aimed at improving the generation of graph-structured data. The author proposed a discrete flow matching framework combined with optimal transport (OT) and an edge-augmented graph transformer. It is the first discrete flow matching generative model with optimal transport for graph data. Additionally, GGFlow introduces a novel goal-guided generation
framework to control the generative trajectory of our model towards desired properties. The experiments show that GGFlow achieves state-of-the-art results in both unconditional and conditional graph and molecule generation tasks.

**Strengths:**

1. This paper is well-written and easy to read.
2. The authors introduced the first discrete flow matching generative model tailored for graph data, leveraging optimal transport for improved efficiency and stability.
3. The performance of the proposed method is quite promising.

**Weaknesses:**

This paper introduces a novel technique that combines discrete flow matching with optimal transport to enhance the efficiency of graph generation. While innovative, borrowing flow matching concepts from diffusion models used in image generation, several aspects could be further clarified:

1. Reward Function Definition: The method relies on a well-defined reward function, which is critical in reinforcement learning (RL) algorithms. However, the paper lacks a discussion on how the choice of reward function impacts the algorithm's performance and outcome.

2. Stability Claims: Although the authors assert that this approach offers greater stability in training and sampling, the experimental results do not provide direct comparisons with other diffusion-based generation algorithms to substantiate this claim.

3. Optimal Transport Justification: The use of optimal transport is presented as an improvement over other flow-matching techniques, yet there is no theoretical evidence provided to explain why optimal transport is preferable in this context. A rationale for this choice would strengthen the paper.

**Questions:**

Although the performance is promising, the optimal transport is quite time-consuming. It is better to provide the complexity analysis or train/inference time compared with other methods.

---

> ### Author Response · Authors · 2024-11-19
>
> We sincerely appreciate the reviewer’s constructive feedback. We have carefully addressed each comment and made the necessary revisions accordingly.
>
> **Q: Reward Function Definition**
>
> A:
> Thank you for your feedback. We acknowledge the importance of the reward function in training. In our initial submission, we used  $|\mu - \mu^*|$  as the reward, aligning with our evaluation metric (MAE). Objectives relevant to the metric can serve as effective rewards to achieve the desired outcomes.
>
> In response to your suggestion, we also implemented  $(\mu - \mu^*)^2$  as a reward function. Our experiments showed that this alternative improved the guidance effect, yielding results comparable to our initial approach.
>
> | Reward Function | mean | variance | validality |
> | --- | --- | --- | --- |
> | $\|\mu - \mu^*\|$ | 0.672 | 0.647 | 92.2 |
> | $(\mu - \mu^*)^2$ | 0.720 | 0.625 | 90.1 |
>
> In many application scenarios, using evaluation metrics or related functions as reward signals can effectively achieve the desired objectives [1, 2, 3]. Our findings align with existing literature.
>
> **Q: Training Stability of GGFlow**
>
> A: We provide training loss and average values on Community-small datasets, comparing our method with DiGress, which shares the same training objectives. For fair comparisons, we analyze GGFlow (w/o both) and GGFlow (w/o Evo) to highlight the advantages of the flow matching framework and optimal transport. These results are detailed in our latest submission (lines 1426-1434) and illustrated in Figure S3 of Appendix J.2.
>
> Our results in Figure S3  indicate that GGFlow (w/o both) and GGFlow (w/o Evo) achieve faster and more effective convergence than DiGress using the same architecture. Furthermore, GGFlow (w/o Evo) surpasses both GGFlow (w/o both) and DiGress in average metrics, demonstrating the benefits of flow matching and optimal transport. These results suggest enhanced stability during training.
>
> Additionally, Figure 3 of our initial submission presents sampling efficiency data, showing that GGFlow has narrower confidence intervals than DiGress and GDSS, indicating improved sampling stability. Our method achieves superior performance with fewer inference steps, further demonstrating its sampling efficiency.
>
> **Q: Optimal Transport Justification**
>
> A: As noted in FM [4], the source and target coupling can be extended to more general cases, and we choose the optimal transport as our coupling. Optimal transport enables our model to pair the nearest source and target data points, ensuring that flow trajectories are optimal [5], which is particularly applicable to graph-structured data. This results in more direct generative paths, enhancing the overall performance of our model.
>
> **Q: Complexity Analysis of Optimal Transport**
>
> A: During inference, we do not need to calculate the optimal transport map as outlined in Equation 6 of our manuscript.
>
> We compared the training times of our method with DiGress using identical architectures on an NVIDIA A100 80G GPU. Considering model size, batch size, and number of nodes, we measured the duration of single training steps across three datasets. Our results show that the time required for optimal transport accounts for only 5% of the total training time, demonstrating its efficiency. This complexity analysis is detailed in our latest submission, specifically in lines 1084-1101 of Appendix E.
>
> |  | DiGress | Training Step | OT Time | Percentage | Model Size | Batch Size | Number of nodes |
> | --- | --- | --- | --- | --- | --- | --- | --- |
> | Planar | 0.1647 | 0.126349 | 0.0024997 | 1.9% | 3.6 M    | 64 | 64 |
> | Zinc250K | 0.1690 | 0.130121 | 0.00699354 | 5.3% | 4.6 M | 128 | [6,38] |
> | Community Small | 0.0456 | 0.04079 | 0.00236716 | 5.6% | 6.4 M | 80 | [12,20] |
>
>
> [1]Liu, Yijing, et al. "Graph diffusion policy optimization." *arXiv preprint arXiv:2402.16302* (2024).
>
> [2] Fan, Ying, et al. "Reinforcement learning for fine-tuning text-to-image diffusion models." *Advances in Neural Information Processing Systems* 36 (2024).
>
> [3] Mnih, Volodymyr, et al. "Human-level control through deep reinforcement learning." *nature* 518.7540 (2015): 529-533.
>
> [4] Gat, Itai, et al. "Discrete Flow Matching." The Thirty-eighth Annual Conference on Neural Information Processing Systems.
>
> [5] Liu, Xingchao, and Chengyue Gong. "Flow Straight and Fast: Learning to Generate and Transfer Data with Rectified Flow." *The Eleventh International Conference on Learning Representations*.

---

> ### Author Response · Authors · 2024-11-22
>
> Dear Reviewer,
> Thank you very much for your thoughtful review. We have addressed the concerns you raised and look forward to your constructive feedback.

---

> > ### Comment · Reviewer_Exo3 · 2024-11-26
> > **Thank you for the response**
> >
> > Thank you for the additional experiments and explanations. The authors have addressed my concerns. I will raise my score to 8.

---

> > > ### Author Response · Authors · 2024-11-26
> > >
> > > We sincerely appreciate the reviewer's insightful and constructive comments, which have significantly enhanced the quality of our manuscript.

---

### Comment · Area_Chair_dCVC · 2024-11-28
**AC's suggestion**

There were notable disagreements about whether the paper meets the acceptance criteria. On the positive side, the results appear promising. However, concerns were raised about the source of the performance improvements. Are these gains genuinely due to the use of flow matching and OT, as suggested in the title, or are they influenced by other factors, such as a better network architecture or the inclusion of additional features? The authors should prioritize addressing these concerns to facilitate constructive discussions between the reviewers and area chairs.

---

> ### Author Response · Authors · 2024-11-29
>
> We will describe our model results from model performance, inference efficiency and training stability to demonstrate that flow matching and optimal transport contribute to our results. Furthermore, we will discuss the issues related to the extra features.
>
> ### Community Small Dataset
>
> | Method          | Deg.  | Clus. | Orbit | Step |
> |-----------------|-------|-------|-------|------|
> | DiGress         | 0.032 | 0.047 | 0.009 | 500  |
> | GGFlow (w/o both)| 0.029 | 0.076 | 0.003 | 500  |
> | GGFlow (w/o OT) | 0.028 | 0.027 | 0.007 | 500  |
> | GGFlow (w/o Evo)| 0.018 | 0.075 | 0.004 | 500  |
> | GGFlow          | 0.001 | 0.084 | 0.004 | 500  |
>
> ### Planar Dataset
>
> | Method          | Deg.    | Clus.   | Orbit  | Spec.  | Val.&Nov.&Uni. | Step |
> |-----------------|---------|---------|--------|--------|----------------|------|
> | DiGress         | 0.0003  | 0.0372  | 0.0098 | 0.0106 | 87.5           | 500  |
> | GGFlow (w/o both)| 0.0023  | 0.1076  | 0.0053 | 0.0099 | 92.5           | 500  |
> | GGFlow (w/o OT) | 0.0015  | 0.0431  | 0.0020 | 0.0067 | 97.0           | 500  |
> | GGFlow (w/o Evo)| 0.0020  | 0.0763  | 0.0034 | 0.0124 | 94.5           | 500  |
> | GGFlow          | 0.0156  | 0.0196  | 0.0019 | 0.0091 | 97.5           | 500  |
>
> Firstly, we provided detailed ablation studies on both the Community Small and Planar datasets. In the settings of our ablation experiments, GGFlow (w/o Evo) and GGFlow (w/o both) share the same architecture which incorporates extra features, and the same hyperparameter settings as DiGress [1]. GGFlow (w/o both) adopts only the flow matching framework without optimal transport, and GGFlow (w/o Evo) incorporates both the flow matching framework and optimal transport.
>
> The results in Table 4 of our submission demonstrate that GGFlow (w/o both) outperforms DiGress across most metrics, highlighting the advantages of the flow matching framework over the diffusion framework in graph space. Moreover, GGFlow (w/o Evo) also outperforms GGFlow (w/o both) in nearly all metrics, indicating the effectiveness of optimal transport. Additionally, according to Figure S2 in Appendix J.2, we also observed that optimal transport enhances the inference efficiency of GGFlow.
>
> Then, we also provided training analysis of using optimal transport in Appendix J.2. According to Figure S3, GGFlow (w/o both) and
> GGFlow (w/o Evo) show better training convergence and stability than DiGress. Moreover, GGFlow (w/o Evo) demonstrates better and more stable generation results than GGFlow (w/o both) during the training procedure, indicating that optimal transport significantly improves the training stability and efficiency of our model.
>
> Finally, regarding the extra features in our model, we conducted experiments on the Zinc250k dataset in our first rebuttal to demonstrate their utility. It is noted that these extra features are not a primary contribution of our models; we clarified in our initial submission that they come from DiGress [1] and also cited DiGress in line 247 of our initial submission. Therefore, we did not provide ablation studies of these extra features in our intial submission. The original DiGress paper provides ablation studies of these features, which can be found in Table 6 of their paper.
> | ZINC250K | Valid | Valid. w/o corr. | NSPDK | FCD |
> | --- | --- | --- | --- | --- |
> | DiGress | 100 | 94.98 | 0.0021 | 3.482 |
> | GGFlow (w/o extra) | 100 | 98.03 | 0.0012 | 1.667 |
> | GGFlow | 100 | 99.63 | 0.0010 | 1.455 |
>
> In summary, we conducted comprehensive ablation studies to demonstrate the effectiveness of our proposed flow matching method and optimal transport. We think these results are sufficient to support the effectiveness of our method and the contributions of our work.
>
> Thank you for your attention and reading.
>
>
> [1] Vignac, Clement, et al. "DiGress: Discrete Denoising diffusion for graph generation." *The Eleventh International Conference on Learning Representations*.

---

### Author Response · Authors · 2024-12-04
**Rebuttal Summary to Reviewers' Feedback**

Dear Reviewers/Area Chairs/Senior Area Chairs,

We sincerely appreciate all reviewers' constructive feedback. We have carefully addressed each comment and made the necessary revisions accordingly. Here are the main revisions in our rebuttal phase:

1. **Optimal Transport:**
   - We revised Theorem 1 to avoid any misleading.
   - We added detailed explanations and rationales about our optimal transport in Appendix C.4 to introduce the plausibility of our proposed OT.
   - We provided a theoretical proof of Lemma 1 in Appendix C.4 to show that our optimal transport preserves the permutation property of generated graphs.
   - We included a time analysis of OT in Appendix E to demonstrate its efficiency.

2. **Ablation Studies:**
   - We added Planar datasets to our ablation studies in Table 4 to further demonstrate the effectiveness of our proposed flow matching method and optimal transport.
   - We provided training and inference analysis of using optimal transport in Appendix J.2 to show that our OT significantly improves the training stability and efficiency of our model.

3. **Experiments:**
   - We provided results on the Enzymes dataset in Appendix H to further demonstrate the effectiveness of our model on large graph datasets.

We appreciate the reviewers' valuable feedback and suggestions, which have significantly improved the quality of our manuscript.

---

### Meta-Review · Area_Chair_dCVC · 2024-12-22

**Metareview:**

The paper introduces a discrete flow matching framework for graph generation, incorporating optimal transport (OT), an edge-augmented transformer architecture (GraphEvo), and goal-conditioned generation. While the integration of discrete flow matching with OT and GraphEvo represents a new development, the submission received mixed reviews. Promising results were noted, but significant concerns were also raised by reviewers.

### Key Concerns:

1. **Fundamental Differences with Diffusion Models:**
   The paper does not clearly articulate the fundamental differences between the proposed discrete flow matching framework and existing discrete diffusion/flow-matching models. It remains unclear why the proposed method outperforms existing graph generation algorithms based on diffusion or flow-matching approaches.

2. **Unclear Novelty and Component Contributions:**
   The submission integrates multiple components, but the specific novelty of each element is ambiguous. If the contribution lies in a particular component, that component should be evaluated within both flow and diffusion models to isolate its impact. Alternatively, if the novelty is in the combination of these components, a detailed theoretical explanation is necessary to justify why this specific combination outperforms alternatives (e.g., substituting flow with diffusion or avoiding pre-ordering the nodes of training and prior graphs).

3. **Permutation-Invariance and Node Ordering Assumptions:**
   The paper’s use of permutation invariance under the assumption of a fixed node order has been a central point of contention. The authors argue that a fixed node order ensures permutation invariance, while a reviewer challenges this claim, questioning how a fixed order can be assumed in permutation-invariant graphs. Despite extensive discussion during the review phase, this critical issue remains unresolved.

4. **Potential Hidden Heuristics Influencing Performance:**
  After synthesizing the reviews and carefully evaluating the paper, the Area Chair has concerns about potential hidden heuristics that may underlie the observed performance gains. Specifically, unanswered questions remain regarding the influence of training data on the prior distribution of nodes and the method by which generated nodes are ordered to align with the training graphs. These aspects are critical for understanding the meaningful application of OT in the context of fixed node orders, as opposed to truly permutation-invariant graphs.

### Summary and Recommendation:
While the paper demonstrates empirical promise, unresolved issues regarding its theoretical foundations, experimental design, and the validity of claims related to OT and permutation invariance undermine its contribution. Additionally, the reliance on pre-existing architectures and features, without substantial advancement in graph generation methodology, renders the novelty incremental rather than transformative. These limitations outweigh the paper's strengths. The authors are encouraged to address these concerns in a revised submission.

**Additional Comments On Reviewer Discussion:**

The discussion phase highlighted ongoing disagreements between the authors and Reviewer PZdL, particularly regarding the role of OT. Although the authors provided clarifications on several points, critical issues regarding OT’s lack of invariance and the unclear novelty of the framework remain unresolved. Reviewer PZdL’s concerns about the compatibility of permutation invariance with fixed node ordering, along with the potentially misleading claims surrounding OT, remain unaddressed.

---

### Decision · Program_Chairs · 2025-01-22

Reject